

# Cloud and DNI nowcasting with MSG/SEVIRI for the optimised operation of concentrating solar power plants

Tobias Sirch[1], Luca Bugliaro[1], Tobias Zinner[2], Matthias Möhrlein[3], and Margarita Vazquez-Navarro[1]

[1]Deutsches Zentrum für Luft- und Raumfahrt, Institut für Physik der Atmosphäre, Oberpfaffenhofen, Germany
[2]Meteorologisches Institut, Ludwig-Maximilians-Universität, Munich, Germany
[3]Nowcast GmbH, Albert-Roßhaupter-Str. 43, Munich, Germany

*Correspondence to:* T. Sirch (tobias.sirch@dlr.de)

**Abstract.** A novel approach for the nowcasting of clouds and Direct Normal Irradiance (DNI) based on the Spinning Enhanced Visible and Infrared Imager (SEVIRI) aboard the geostationary Meteosat Second Generation (MSG) satellite is presented for a forecast horizon up to 120 min. The basis of the algorithm is an optical flow method to derive cloud motion vectors for all cloudy pixels. To facilitate forecasts over a relevant time period, a classification of clouds into objects and a weighted triangu-
lar interpolation of clear sky regions is used. Low and high level clouds are forecasted separately because they show different velocities and motion directions. Additionally a distinction in advective and convective clouds together with an intensity correction for quickly thinning convective clouds is integrated. The DNI is calculated from the forecasted optical thickness of the low and high level clouds. In order to quantitatively assess the performance of the algorithm, a forecast validation against MSG/SEVIRI observations is performed for a period of two months. Error rates and Hanssen-Kuiper skill scores are derived
for forecasted cloud masks. For a forecast of 5 min for most cloud situations more than 95% of all pixels are predicted correctly cloudy or clear. This number decreases towards the 2 h forecast towards 80-95% depending on cloud type and vertical cloud level. Hanssen-Kuiper skill scores for cloud mask go down to 0.6-0.7 for a two hour forecast. Compared to persistence an improvement of forecast horizon by a factor of two is reached for all forecast up to 2 h. Comparison of forecasted optical thickness distributions and direct normal irradiance at ground against observations yields correlation coefficients larger than
0.9 for 15 min forecasts and around at least 0.65 for 2 h forecasts.

## 1 Introduction

Availability of energy power plays a central role for society and their economical evolution. Among the renewable energy sources, concentrating solar power (CSP) systems have a great potential since they combine electricity production with a storage capacity. By means of mirrors the incoming solar irradiance is concentrated, heating a fluid and driving a heat engine. The used technologies are parabolic trough, solar power tower, Fresnel reflectors and dish Stirling. In case of low insolation the electricity production is taken over by a fuel, e.g. gas. The operation of such solar power plants is challenging since the thermodynamic properties of the heated fluid are difficult to control in case, for instance, when the CSP plant is only partially illuminated by the Sun or when insolation is strongly variable over time ranges of a few minutes to a few hours.





The fuel of solar power plants is direct normal irradiance (DNI). The main source of its spatio-temporal variability is cloudiness due to their intrinsic spatio-temporal inhomogeneity and to the fact that already thin clouds can reduce DNI to unusable levels for CSP. Further factors that affect DNI are aerosols and, to a lesser extent, water vapour and ozone (Gueymard, 2012). Thus, the knowledge and the prediction of cloud properties for the derivation of DNI is essential for the optimisation of the

CSP operation strategy as for day-ahead electricity markets the electricity production must be announced to the market operator and deviations from the production schedule may lead to deviation penalties. Kraas et al. (2013) show the economic merits of a forecasting system for day-ahead forecasts for concentrated solar power, which reduces penalties by 47.6% compared to a persistence model. The persistence model is the simplest forecast model and works well for periods of low cloud variability, obviously for clear-sky cases. Of course, the accuracy of persistence models is reduced substantially when the

variability increases. Therefore other methods are used based on data from various sources depending on the forecast horizon. For the prediction of solar irradiance one or two days ahead numerical weather prediction (NWP) models are used (Marquez and Coimbra, 2011), which provide better results if combined with artificial neural networks (ANNs) (Gonzalez et al., 2010). However, for short-term forecasts up to 6 h NWP models are not well suited and satellite based methods come into play: Perez et al. (2013) and Lorenz et al. (2011) show that below a forecast horizon of 5-6 h forecasting methods of NWP models have a

lower accuracy compared to satellite based algorithms (Eissa et al., 2013; Marquez and Coimbra, 2013). Other approaches deal with the detection and tracking of cloud patterns with satellite data - a challenging task due to the non-linearity in atmospheric motion. They range from standard pattern recognition techniques (Bolliger et al., 2003; Wu, 1995; Wu et al., 1997; Schmetz et al., 1993a) to multichannel correlation-relaxation labeling (Evans, 2006). Geostationary satellites usually provide multispectral images with a temporal resolution between 5 and 30 min that can be used for determining the motion of clouds and their

properties. However, their temporal and spatial resolution (in the order of kilometers) of satellite images are inappropriate for accurate forecasts of clouds for the next few minutes at particular (power plant) sites. This lack can be overcome by using local high-frequency image-capturing devices, such as Total Sky Imagers (Chu et al., 2013; Wacker et al., 2015), or other ground sensors (e.g. pyranometers, Bosch et al. (2013)).

Convective clouds are of particular interest for society due to the high precipitation rates that are often connected to them.

Because of their rapid development they increase the error in any forecast. Therefore it is reasonable to treat convective and advective clouds separately and to investigate the development of convective cells, which is a challenging task. During the last decades a large number of different cloud nowcasting approaches has been developed, most of them with a strong focus on thunderstorms. These techniques are using near real-time information from radars, e.g. CONRAD (Lang, 2001) and TRT (Hering et al., 2004), and passive imagers, e.g. Bolliger et al. (2003), Zinner et al. (2008) or Feidas and Cartarlis (2005), or a

combination of both (Henken and Schmeits, 2011). There is also an effort to combine the advantages of radar data with lightning data (Steinacker et al., 2000) or additional numerical models (Pierce et al., 2000), but these methods have the disadvantage, that only these areas can be observed, where the used instruments are sited.

This publication presents a novel nowcasting algorithm based on satellite data from the SEVIRI imager aboard Meteosat Second Generation (MSG) for all clouds with a focus on clouds which are relevant for CSP generation. With its high repetition

rate of 15 min, its spatial sampling distance of 3 km and the availability of 12 spectral channels, this sensor is very well suited



for the determination and forecast of cloud optical properties to be used to derive DNI since clouds are highly variable in space and time. Our method focus on forecast times from 5 to 120 min. It exploits an optical flow algorithm to determine atmospheric motion vectors for every pixel. The starting point is represented by the optical thickness of clouds that are first split up into two (vertically overlapping) layers in order to take care of different velocities of upper level and low level clouds.

To reduce the turbulent character of the atmospheric motion field on small scales, rendering long range forecasts impossible, cloud subsets are defined as rigid objects that move with time. Convection cannot be forecasted adequately this way, but our approach considers dissipating convective clouds, where extended anvils are produced that can live for many hours and have an important impact on DNI. DNI itself is eventually computed from the cloud optical thickness forecast. A validation against MSG/SEVIRI observation is shown at the end.

After a description of the satellite instrument MSG/SEVIRI and the cloud detection and analysis algorithms needed for our forecast method including the optical flow procedure (Sect. 2), the nowcasting algorithm is presented in Sect. 3 and its validation in Sect. 4.

## 2   Instrument and Tools

Meteosat Second Generation (MSG) is a series of European geostationary satellites operated by EUMETSAT. Their primary
mission is the observation and forecasting of weather phenomena on the earth's full disk. For this purpose the 12-channel imager SEVIRI (Spinning Enhanced Visible and Infrared Imager) has been developed. Table 1 shows its spectral channel characteristics consisting of three channels in the visible, one in the near infrared and eight in the infrared spectral range with a sampling distance of 3 km at the sub-satellite point (Schmetz et al., 2002). Additionally a broadband high resolution visible (HRV) channel is integrated, which covers only a part of the earth's full disk with a higher spatial resolution of 1 km at the
sub-satellite point. The usual repetition rate of 15 min is reduced to 5 min in rapid-scan-mode for a part of the disk.

Optical, micro- and macrophysical properties of clouds are important parameters for the modeling of radiation-cloud inter-actions. Thus, their determination plays an important role for the computation of surface radiation, including direct normal irradiance (DNI). The cloud detection and analysis algorithms used in this work are presented in the following two sections (Sect. 2.1, 2.2). In Sect. 2.3 an optical flow method for the determination of cloud motion fields is described.

**2.1   Cloud Classification and Cloud Optical Properties**

For thin ice clouds The "Cirrus Optical properties derived from CALIOP and SEVIRI during day and night" (COCS, Kox et al. (2014)) algorithm is used. It is a backpropagation neural network, which is trained with collocated products of the depolarization-lidar CALIOP (Cloud-Aerosol Lidar with Orthogonal Polarization) aboard the Cloud-Aerosol Lidar and In-frared Pathfinder Satellite Observations (CALIPSO) and brightness temperatures as well as brightness temperature differences
of MSG/SEVIRI (see above). The COCS algorithm provides optical thickness and cloud top height for ice clouds. In a valida-tion study against airborne High Spectral Resolution Lidar measurements (Kox et al., 2014), COCS detected 80% of the cirrus clouds with optical thickness 0.2 and its detection efficiency increased for higher optical thicknesses. For optical thickness 0.1





COCS detected still 50% of the cirrus clouds observerd by CALIPSO. The false alarm ratio amounted to 2.6% for all measured cirrus clouds. It is very robust for small optical thicknesses above 0.1 up to a maximum of 2.5. Clouds with larger optical thickness cannot be penetrated by CALIOP (Winker et al., 2010). This also means that COCS provides information only about the highest ice cloud layer. As COCS works with the thermal SEVIRI channels it can be applied during day and night.

In addition to COCS, APICS ("Algorithm for the Physical Investigation of Clouds with SEVIRI", Bugliaro et al. (2011)) is applied for the detection of liquid water clouds and thick cirrus. The APICS cloud detection is based on Kriebel et al. (2003) and consists of two groups of threshold tests consisting in reflectivity tests and spatial coherence tests applied to the solar SEVIRI channels. The first group detects a cloud if it is bright enough compared to the cloud free reflectivity. The second is applied over sea and detects a cloud if the variability of the signal is higher than that of the cloud free background (the sea

surface reflectivity is supposed to be spatially homogeneous). A pixel is cloudy if at least one test gives a positive result.

A cloud top phase mask is produced by merging the results of the two algorithms. A cloud detected by COCS is an ice cloud, i.e. its cloud top is composed of ice. Clouds detected by APICS can be both ice and liquid water clouds since APICS detects clouds that are bright enough or whose spatial variability is high enough. Thus, we assign all clouds detected by COCS to the ice phase, while a cloud that is detected by APICS but not by COCS is assigned to the liquid water phase.

In a second step, cloud optical properties are obtained the following way. COCS provides the optical thickness of the upmost cloud layer. Furthermore, for both cloud types (liquid water and ice), cloud optical thickness and effective radius are derived from two solar channels using APICS. In APICS, SEVIRI channels centred at 0.6 and 1.6 $\mu$m are used based on the method by Nakajima and King (1990); Nakajima and Nakajima (1995): in the first spectral range clouds mainly scatter radiation, while in the second channel clouds both absorb and scatter the incoming solar radiation. Since absorption is a function of effective

radius while scattering depends on optical thickness, a simultaneous retrieval of optical thickness and effective radius can be performed by minimizing the difference between measured and computed reflectivities. The look-up tables required for this application have been computed with the radiative transfer model libRadtran (Mayer and Kylling, 2005) using a mid-latitude standard atmosphere (Anderson et al., 1986), a typical continental aerosol load (a rural type aerosol in the boundary layer, background aerosol above 2 km, spring-summer conditions and a visibility of 50 km, Shettle (1989)). Surface albedo is taken

from the temporally appropriate white sky MODIS albedo product MCD43C1 (Schaaf et al., 2002). For liquid water clouds, Mie cloud optical properties are assumed (spherical particles), while for ice clouds the parametrization by Baum et al. (2005) is used (a mixture of ice cloud habits as a function of ice crystal size). The optical thickness derived this way ranges from 0 to 100 and refers to the entire atmospheric column.

Thus, for ice clouds two sets of optical thickness values are obtained, one using COCS and one using APICS. The main

difference consists in the fact that APICS, in contrast to COCS, derives the cloud optical thickness of the entire atmospheric column and not only of the upmost ice cloud layer. For this reason, optical thickness from APICS is usually higher than from COCS.



## 2.2 Quickly Developing Convective Clouds

For the detection of convective clouds, methods of the Cb-TRAM algorithm (CumulonimBus TRacking And Monitoring, Zinner et al. (2008, 2013)) have been examined and exploited in this work. Cb-TRAM divides convection into three stages:

- convection initiation (stage 1)

- rapid cooling (stage 2)

- mature thunderstorm cells (stage 3)

The detection of convection initiation (stage 1, Merk and Zinner (2013)) uses the evolution of reflectance in the HRV and cooling rate in the thermal spectral range. Stage 2 is issued for rapid vertical developments detectable in the water vapour channels at $6.2\,\mu$m. The detection of mature thunderstorm cells (stage 3) is limited to areas with a strong spatial roughness of the HRV, determined by the local standard deviation, combined with the brightness temperature difference of $6.2\,\mu$m and $10.8\,\mu$m. This is only valid for daytime. During night the HRV is replaced by a similar measure for the brightness temperature at $6.2\,\mu$m which is less successful at confining the detection to the active updraft, but includes large parts of the surrounding anvil and is independent of sunlight conditions, i.e. it produces smooth results in particular at sunrise, when the forecast is started. As these thinner clouds are most interesting for the derivation of DNI, the latter detection is used as part of the forecast algorithm in the following.

## 2.3 Cloud Motion Fields

The forecast rests upon an optical flow method determining a motion vector field from two consecutive images which is part of Cb-TRAM (Zinner et al. (2008)). Unlike feature based approaches, often used for the determination of atmospheric motion vectors for single objects/clouds in an image (e.g. Schmetz et al., 1993b), this method is pixel-based: instead of vectors only for interesting cloud patterns a disparity vector field $\overrightarrow{V}(P)$ defined at each pixel position $P$ is derived.

Movements in the atmosphere take place on different scales reaching from microscale (few cm) to global scale ($10000\,$km). These large-scale flows overlay the small-scale movements so that the determination of the disparity vector field for all scales is challenging. In order to take this into account the disparity vector fields are successively derived on different scales, starting from low resolution down to high resolution - a pyramidal scheme.

The procedure is described by means of an example for two images $A$ and $B$ (Fig. 1a,b) with a size of $nx = 100 \times ny = 100$ pixels displaying two squares ($10 \times 10$ pixels). Similar structures in the images $A$ and $B$ are identified iteratively at different spatial scales, i.e. for all sub-sampling levels $l$ of the pyramidal approach with $M$ levels, starting with the topmost level with the roughest resolution:

1) selection of the number of sub-sampling levels $N$ (e.g. $N = 2$ for a pyramid with $M = N + 1 = 3$ levels): this number depends on the size of the shifts that are expected

2) definition of the image $A_M = A$ and set $l = M - 1$




3) Start the iterative process

3.1) calculation of dimensions $nx_l = nx/2^l$ and $ny_l = ny/2^l$ of the given sub-sampling level $l$ ($nx_l \times ny_l = 25 \times 25$ pixels for the topmost level, $nx_l \times ny_l = 50 \times 50$ pixels for the second sub-level)

3.2) resampling of the start images $A_{l+1}$ and $B$ to $nx_l \times ny_l$ to obtain $A_l$ and $B_l$

3.3) determination of comparison images $A_{l,s}$ by shifting every pixel $P$ in image $A_l$ to $A_{l,s} = A_l(P + \overrightarrow{\Delta K}_{i,j})$ by $\overrightarrow{\Delta K}_{i,j} = (i,j)$, $i,j \in [-2,2]$, in both dimensions

3.4) identification of the best fit between all possible $A_{l,s}$ and the target image $B_l$ through minimisation of the squared difference of the intensities of $A_{l,s}$ and $B_l$ in a surrounding of each pixel: this results in the disparity vector field $\overrightarrow{V_l}$ with dimensions $nx_l \times ny_l$

3.5) to mitigate of the impact of singular incorrect motion derivations and ensure physically realistic local flow fields, these initially integer displacements $\overrightarrow{V_l}$ are smoothed over the local neighbourhood with the same Gauss kernel as above

3.6) blowing up the resolution of $\overrightarrow{V_l}$ to the original one $nx \times ny$ to obtain $\overrightarrow{V}_{l,original}$

3.7) adding the motion vectors obtained so far to $\overrightarrow{V} = \sum_{i=l}^{N} \overrightarrow{V}_{l,original}$

3.8) warp the image $A$ with the disparity vector field $\overrightarrow{V}$ to

$$A_l(P) = A(P - \overrightarrow{V}(P)) , \tag{1}$$

for every pixel position $P$. Notice that this equation implies that pixels $P$ in $A$ are not shifted with $\overrightarrow{V}$ into $A_l$, but for every pixel $P$ in the forecast image $A_l$ a value from the starting image $A$ is assigned which can be found there at position $P - \overrightarrow{V}$. So every pixel is allocated to a value and no information gaps (i.e. no "holes" in the image $A_l$) occur in the forecast. However, since $\overrightarrow{V}$ contains floating point values due to the smoothing in step 3.5, bilinear interpolation of $A$ in x and y is applied when performing Eq. 1. Thus, the warped image is only a "remapping" of the start image and tmporal evolution trends of image values can be recognised and implemented.

3.9) reduce the value of $l$ by 1 and go back to step 3.1 if $l \geq 0$.

4) End of the iterative procedure: The refined disparity vector field $\overrightarrow{V}$ that has been obtained through successive addition of the results of all pyramidal levels provides the final disparity vector field $\overrightarrow{V}_{A \to B}$ in full resolution and its application to Eq. 1 the final warped image $A'(P) = A_{l=0}(P) = A(P - \overrightarrow{V}(P))$.

The refined disparity vector field $\overrightarrow{V}_{A \to B}$ in full resolution is shown in Fig. 1c. Notice that the it is different from zero not only over the area defined by the initial image $A$ but also in the direct neighbourhood. Due to that, disparity vectors are not always pointing from $A$ to $B$ but, outside of image $A$ and $B$, also in other directions. Figure 1c also shows the final warped





image $A'$. The displacement of image $A$ onto $B$ shows good results as the final remaining difference field (Fig. 1d) $A' - B$ exhibits only small differences on the edges of the squares caused by the relaxation of the disparity analysis by smoothing.

For more details, technicalities and an additional example please see Zinner et al. (2008).

## 3 Forecast Algorithm

In this section the forecast algorithm is described. It exploits the methods introduced in the previous section. First, a more advanced cloud classification is presented that distinguishes two overlapping classes of clouds. Then, the pixel-based disparity vector field is determined for both cloud classes. Cloud objects are formed, based on optical thickness, and motion vectors are derived for these objects. After the assignment of motion vectors to cloud free areas, clouds are warped to their new position with this motion vector field. An intensity correction is applied for rapidly thinning convective clouds. In a last step the direct

normal irradiance (DNI) is calculated from the optical thickness.

### 3.1 Step 1: Cloud Classification

In the following clouds are classified in MSG/SEVIRI images according to two criteria: The first one considers the cloud top phase and the vertical structure of clouds (Sect. 3.1.1), the second identifies a type of convective clouds particularly relevant for our application, dissipating convective clouds with a thinning anvil (Sect. 3.1.2).

### 3.1.1 Cloud Optical Properties

Low level and high level clouds are often observed to move in different directions at different velocities due to complex wind profiles in the atmosphere. In order to take this aspect into account, we aim at the separation of low and high level clouds and the generation of two forecasts, one for low level and one for high level clouds. However, using APICS and COCS applied to MSG/SEVIRI satellite data according to Sect. 2.1, this is only possible to some extent. A high ice cloud layer as detected

by COCS might occur in the same pixel as a low level liquid water cloud. Optical thickness of the cirrus cloud is then well accounted for by the COCS result, while APICS provides an approximation of the total optical thickness of the upper ice cloud and the lower water cloud together. Inaccuracies are due to the fact that cloud optical thickness is always derived by APICS according to the given cloud top phase, when the atmospheric column consists of both liquid water and ice this assumption fails and the resulting optical thickness is only an approximation to the correct total optical thickness. Furthermore, the ice layer

detected by COCS might be the upper layer of a vertically and optically much thicker cloud like a Cb (Cumulonimbus). In this case, the total optical thickness of the cloud is most likely much larger than the COCS maximum value of 2.5, and APICS can much better capture this aspect since its optical thickness is based on the reflectivity of the entire atmospheric column.

In general, the discrimination among all these cases and the determination of optical properties for all cloud layers is challenging using only passive satellite observations. Several approaches have been proposed, for instance (Joiner et al., 2010;

Gonzalez et al., 2003; Huang et al., 2005; Baum et al., 1995). In this work, we want to forecast surface DNI, which becomes one per mill of the original value for a slant optical thickness of 7. To this end, all liquid water clouds usually reduce DNI to





values far below the range interesting for CSP production due to their high optical thickness. Thus, accuracy in this range is not crucial. In case of thin cirrus, however, surface DNI is not zero and the accuracy of the ice cloud optical thickness is important as CSP, like parabolic troughs, is shut down when DNI $< 200\,\mathrm{W\,m^{-2}}$ , which corresponds to a vertical optical thickness of 2.

Qualitative indications contained in the false colour composites can be exploited to provide a reasonable differentiation

between one layer and two layer cloud situations. In the false colour composites, the vertical structure of the clouds is suggested by the colours: a yellow component is always associated with low (warm) clouds (i.e. a small blue component for which the inverted IR_108 channel is used), while bluish or violet clouds are produced by low temperatures (i.e. high blue contributions). For clarification an example is depicted in Fig. 2 for 7 April 2013. On this day a frontal zone is crossing the Iberian Peninsula. The different cloud types are illustrated by a SEVIRI false colour composite (Fig. 2a). The yellow colored cloud of the frontal

zone consist of low, warm water clouds. In other regions these clouds are overlaid by high thin ice clouds (blueish colors in Fig. 2a). Over the Mediterranean Sea Eastern of Gibraltar sngle layer ice clouds are observed. We exploit the differences between the APICS and COCS results for ice and liquid optical thickness to define two classes of clouds called upper clouds and lower clouds that enable us to differentiate among these cases. These two cloud classes can overlap. The classification is summarised in Table 2 and explained in detail in the following.

Liquid water clouds identified following Sect. 2.1 are assigned to the lower cloud layer and their optical thickness is the APICS optical thickness. If APICS and COCS indicate a thin ice cloud ($\tau_{APICS,ice} \leq 2.5$), the presence of an ice cloud without lower liquid cloud layers is assumed and optical thickness of COCS is assigned to the upper cloud layer (because COCS is assumed to be more accurate than APICS for thin cirrus clouds). For a cloud with ice top and APICS ice optical thickness larger than 2.5, the difference between the APICS ice optical thickness and the COCS optical thickness is investigated. If

their difference is smaller than 2.3 ($\tau_{APICS,ice} - \tau_{COCS} \leq 2.3$), this is interpreted as possible deviation between two different methods providing results for the same cloud: please notice that if $\tau_{APICS,ice}$ is smaller than $\tau_{COCS}$ then this condition is always fulfilled, while an upper limit to $\tau_{APICS,ice}$ is set here as $\tau_{COCS} + 2.3$. Even if the situation of a thin liquid water cloud (included in APICS and not captured by COCS) cannot be excluded a priori, we assume that this is not the case here because there is no clear indication for this in the data. Since COCS is supposed to be more accurate than APICS for thin cirrus (and

for the sake of a "consistent" treatment of thin ice clouds in this paper) COCS optical thickness is selected for the upper cloud while the lower cloud optical thickness is set to zero. When in contrast APICS retrieves an ice optical thickness larger than 2.5 (the upper limit of COCS) and the difference between APICS and COCS is larger than 2.3 ($\tau_{APICS,ice} - \tau_{COCS} > 2.3$), this difference is assumed to have physical reasons due to the presence of a lower cloud layer. The situation encountered here is thus either a thin ice cloud on top of a water cloud, or a vertically extended cloud with ice, liquid or mixed-phase microphysics

below the upper ice layer. In all cases the cloud is thick enough that DNI at the surface is diminished to below 10% of its TOA value. Anyway, the correct optical thickness distribution between lower and upper cloud cannot be determined. This problem is solved in the following way: The optical thickness of the lower cloud is set to $\tau_{APICS,ice}$ and that of the upper cloud is set to $\tau_{COCS}$. This ensures that the upper cloud is considered correctly in case of a thin cirrus on top of a low liquid water cloud. Then, when the upper and the lower clouds are moving in different directions and the sun can shine through the thin cirrus to the

ground, the most appropriate ice cloud optical thickness is used. A high accuracy of the liquid water optical thickness cannot



be achieved: it would require a retrieval exploiting solar channels with two cloud layers, a liquid water cloud below and an ice cloud above, i.e. with four unknown variables, optical thickness and effective radius of both layers, or at least three unknown variables if the ice cloud optical thickness derived by COCS using thermal channels is taken for granted and inserted into this imaginary solar retrieval. However, liquid cloud optical thickness is not important because it is usually so high that no DNI

can illuminate the surface. This decision provides an arbitrary ice optical thickness assignment with respect to the upper and lower layers in the case of a vertically extended, Cb-like cloud. Nevertheless, this is again not crucial for our application since the cloud usually moves as a whole (i.e. lower and upper layer continue to overlap) and its total optical thickness is so high that DNI at the surface is always zero. The case where $\tau_{APICS,liq} > 0$, $\tau_{APICS,ice} = 0$ und $\tau_{COCS} > 0$ cannot occur since it is not foreseen in the cloud top phase mask described in Sect. 3.1.1 which builds the starting point of the present classification.

For this classification, the threshold value of 2.3 used above has been determined empirically based on visual inspection of false colour composites like the one shown in Fig. 2 since the real vertical structure of the clouds and the real optical thickness of the cloud layers cannot be derived quantitatively from MSG/SEVIRI observations. The value of 2.3 ensures that when the COCS optical thickness is close to its upper limit of 2.5 the APICS optical thickness must be almost twice as large in order to indicate a multi-layer cloud situation. This classification does not claim to be exhaustive and could be further optimised e.g.

by the use of CALIPSO/CALIOP lidar data (Winker et al., 2009). However, it has the advantage of being computationally fast since it does not require to apply an additional cloud optical thickness retrieval in the case of multi-layer clouds and enables to take care at least partially of the cases when a thin ice cloud is found on top of low level clouds that often move into different directions such that the extrapolation has the possibility to account for this.

Summarising, the cloud classification presented above provides two possibly overlapping cloud layers: the lower cloud layer

with an optical thickness between 0 and 100 and the upper cloud layer with an optical thickness between 0.1 and 2.5. Even if this method is not perfectly accurate, at least, it enables the detection of liquid water clouds below thin ice clouds and the discrimination between thin and thick ice clouds with a theoretical positive impact on the accuracy of DNI. The fact that lower water clouds can be observed simultaneously to ice clouds represents a great advantage for the tracking of low clouds. It is now possible to follow such a cloud when it is shaded by the advection of a thin cirrus cloud as long as the cirrus is thin enough.

The results of the classification applied to Fig. 2a are provided in Fig. 2b,c,d . The yellow colored clouds (low warm clouds) are characterised by the blue colored region in the cloud phase mask shown in Fig. 2b, unless they are overlaid by high thin ice clouds (blueish colors in Fig. 2a), in which case they are detected as multi-phase clouds in the cloud mask (red color). The green areas in Fig. 2b denote single-layer ice clouds. Cloud free areas are depicted in white. Fig. 2c and Fig. 2d show the corresponding optical thickness for the upper and lower cloud layer derived by COCS and APICS according to the prodecure

described above and in Tab. 2.

### 3.1.2 Convective Clouds

The focus of the presented forecast method is laid upon the accurate prediction of thin ice clouds since they modulate surface DNI in the relevant range. Often ice clouds are formed by convection. In contrast to most ice clouds that are mainly characterised by horizontal advection, convective clouds show a strong local vertical development. While during growth and maturity





of convective cells large optical thickness values dominate and DNI at surface is negligible, anvil ice clouds formed during maturity can live much longer than the thunderstorm cloud itself during the decaying stage (Byers and Braham, 1948). Thus they can lead to large but isolated cirrus clouds that are indeed interesting for the DNI forecast at the surface. Considering that convection is stronger and more important at low latitudes, where the solar power potential is high too, the specific consideration

of decaying convective clouds represents an important aspect.

For this reason, a third class of clouds is defined: we single out mature convective clouds using the stage 3 detection of the Cb-TRAM algorithm as discussed in Sect. 2.2. This classification is independent of the previous classification in lower and upper cloud layers (Sect. 3.1.1), but due to the nature of the convective life-cycle Cb-TRAM stage 3 detections turn out to always belong to the upper cloud layer.

## 10   3.2   Step 2: Motion Vectors

Once clouds have been classified and cloud optical thickness has been determined (Sect. 3.1.1), lower and upper clouds can be considered separately, i.e. two forecasts are implemented, one for lower and one for upper clouds. This separation is necessary since motion vectors of low and high clouds differ in most cases because of the different dynamics in these atmospheric layers. In particular, the wind speed in the troposphere usually exhibits very strong variations with altitude. Thus we proceed in the

following way: First, the matcher described in Sect. 2.3 is applied separately to lower and upper clouds. Thus, it produces two independent "pixel-based" motion vector fields (Sect. 3.2.1). Second, for reasons that will become evident below, these two motion vector fields are averaged over specific cloud subsets (Sect. 3.2.2). Finally, motion vectors are provided for the cloud free areas (Sect. 3.2.3).

### 3.2.1   Pixel-based Motion Vectors

In this first stage motion vector fields are derived for the optical thickness of lower clouds and upper clouds separately. Since convective clouds as defined in Sect. 3.1.2 are a subset of the upper clouds, they are not mentioned explicitly here since they do not play any role at this point. Optical thickness of lower clouds attains values from 0 to 100. For upper clouds the range is $[0.1, 2.5]$. Remark that in order to avoid edge effects one should match areas larger than the given region of interest: the area used should be as large as to allow the observations of all clouds that will enter the region of interest during the time span

needed for the forecast, in this case 120 min.

There are two reasons for the use of the optical thickness as input parameter for the matcher: first, it is the quantity which is needed for the calculation of DNI (see Sect. 3.5). Second, the matcher works best if only objects that are actually moving are matched against each other - in this case the cloud objects.

Forecasts are produced in forecast steps of $\Delta t_f = 5$ min up to a forecast horizon of 120 min. First, the disparity vector field

$\overrightarrow{V}_{A \rightarrow B}$ between the intial images $A$ and $B$ separated by a time interval $\Delta t = 15$ min is determined by the pyramidal matcher with $N = 3$ pyramidal sampling levels (see Sect. 2.3). Accordingly, the possible "search radius" is given by at least $2^{(N+2)} = 32$ pixels corresponding to an atmospheric motion of more than 360 km/h at mid-latitudes for the operational MSG/SEVIRI scan mode with 15 min repetition time (see also Zinner et al. (2008)). Then, a disparity vector field $\overrightarrow{V}_{5min}$ according to the length





of the time step $\Delta t_f = 5\,\text{min}$ is computed by multiplication of the disparity vector field $\overrightarrow{V}_{A \to B}$ by

$$d = \Delta t_f / \Delta t \,,$$

i.e., by multiplication with the factor $d = 1/3$:

$$\overrightarrow{V}_{5\,min} = d \cdot \overrightarrow{V}_{A \to B} \,.$$

The forecast image $F_{5\,min}$ for the lead time of 5 min is then produced according to Eq. 1 with the corresponding disparity vector field $\overrightarrow{V}_{5\,min}$ applied to the later initial image $B$:

$$F_{5\,min}(P) = B(P - \overrightarrow{V}_{5\,min}(P)) \ \text{ for all pixels } P \,.$$

Forecasts with longer lead times can be performed as well. For the next forecast step of 10 min the 2 dimensional disparity vector field $\overrightarrow{V}_{5\,min} = (u_{5\,min}, v_{5\,min})$ is shifted with itself. Thereby, the components of the motion vector are advected according

to the cloud/air motion:

$$u_{5\,min,shifted}(P) = u_{5\,min}(P - \overrightarrow{V}_{5\,min}(P))$$

$$v_{5\,min,shifted}(P) = v_{5\,min}(P - \overrightarrow{V}_{5\,min}(P)) \,.$$

The shifted disparity vector field $\overrightarrow{V}_{5\,min,shifted} = (u_{5\,min,shifted}, v_{5\,min,shifted})$ provides the information about the disparity vector field at the position where the pixels will be located according to the atmospheric flow after 5 min. This vector is then

added to $\overrightarrow{V}_{5\,min}$ to produce $\overrightarrow{V}_{10\,min}$

$$\overrightarrow{V}_{10\,min}(P) = \overrightarrow{V}_{5\,min}(P) + \overrightarrow{V}_{5\,min}(P - \overrightarrow{V}_{5\,min}(P))$$
$$= \overrightarrow{V}_{5\,min}(P) + \overrightarrow{s}^{(1)}(P) \,.$$

The first term on the right hand side represents the displacement during the first 5 min, while the second term

$$\overrightarrow{s}^{(1)}(P) := \overrightarrow{V}_{5\,min}(P - \overrightarrow{V}_{5\,min}(P)) = \overrightarrow{V}_{5\,min,shifted}(P)$$

describes the displacement during the 5 min after the initial 5 min time step. The forecast image $F_{10\,min}$ for the lead time of 10 min is thus

$$F_{10\,min}(P) = B(P - \overrightarrow{V}_{10\,min}(P)) \ \text{ for all pixels } P \,.$$

The disparity vector field for 15 min can be expressed as

$$\overrightarrow{V}_{15\,min}(P) = \overrightarrow{V}_{10\,min}(P) + \overrightarrow{s}^{(2)}(P - \overrightarrow{s}^{(2)}(P)) \,,$$

where $\overrightarrow{s}^{(2)}(P) = \overrightarrow{s}^{(1)}(P - \overrightarrow{s}^{(1)}(P))$ describes the displacement in the 5 min after the first two time steps, i.e. after the initial 10 min. This procedure is iterated for further time steps according to the general formula

$$\overrightarrow{V}_{n \cdot 5\,min}(P) = \overrightarrow{V}_{(n-1) \cdot 5\,min}(P) + \overrightarrow{s}^{(n-1)}(P - \overrightarrow{s}^{(n-1)}(P))$$
$$F_{n \cdot 5\,min}(P) = B(P - \overrightarrow{V}_{n \cdot 5\,min}(P)) \,, \ \ n \geq 2 \,, \tag{2}$$





where any $\overrightarrow{s}^{(n)}$ is determined recursively as

$$\overrightarrow{s}^{(n)}(P) = \overrightarrow{s}^{(n-1)}(P - \overrightarrow{s}^{(n-1)})$$
$$\overrightarrow{s}^{(1)}(P) := \overrightarrow{V}_{5\,min}(P - \overrightarrow{V}_{5\,min}(P)) \,.$$

Physically, this approach means that the motion vector field $\overrightarrow{V}_{5\,min}$ is supposed to describe the atmospheric flow as it can

de determined from the two initial images. The forecast procedure, Eq. 2, follows the atmospheric flow in steps of 5 min by evaluating $\overrightarrow{s}^{(n)}$ at the different positions a cloud/air parcel runs through with time.

To illustrate the result of this forecast procedure we consider the upper cloud layer from the example in Fig. 2. The optical thickness of these clouds is depicted in Fig. 3 for 13:00 UTC (a) and 13:15 UTC (c). The disparity vector field $\overrightarrow{V} = \overrightarrow{V}_{A \rightarrow B}$ obtained from these two images is also displayed in Fig. 3c using small arrows. For clarity only one out of ten vectors is shown.

Nonetheless already this way a very large motion vector variability is visible, especially inside cloud regions and close to them. The large cloud field in the eastern part of the Iberian Peninsula is generally shifted towards East or North-East. However, motion vectors abruptly vary from one pixel to the next. The full motion vector field is applied to the 13:15 UTC image (using Eq. 2) to produce a 1 h-forecast (Fig. 3d). This forecast shows several deficiencies compared to the real cloud optical thickness observed at this time (Fig. 3b) for the following reason: The pyramidal matcher provides a detailed motion field representative

only for changes during a (short) 15 min time period. Small scale turbulence and changes produce a very variable disparity vector field (in direction and absolute value) not representative for a longer time period. As a consequence cloud patterns dissolve into small patches within a short period of time, which does not correspond to reality as only the average larger scale motions stay stable over longer periods.

### 3.2.2 Object based Cloud Motion Vectors

For the reason discussed above, an averaging procedure for the pixel-based cloud motion vectors is implemented. To this end, neighboring pixels with similar cloud characteristics (here optical thickness) are combined to objects. This procedure is called object classification and is applied separately to upper and lower cloud layers since they are forecasted separately. At this step, convective clouds are treated separately. This averaging procedure removes small-scale variability which is realistic at the moment of derivation, but makes the forecast unstable.

For upper clouds, first each convective cell (Sect. 2.2) is classified as an individual object as prerequisite for the application of a specific procedure presented further down (Sect. 3.3). For the remaining part of the upper cloud layer, the optical thickness range $[0.1, 2.5]$ is divided into eight classes with a bin size of 0.3 to create objects. An example for this object classification is depicted in Fig. 4. The upper cloud layer (left panel in Fig. 4), that does not contain any convective cell in this case, is separated into 39 objects (right panel in Fig. 4). Each object consists of all contiguous pixels belonging to the same of the eight optical

thickness classes. The size of the single objects varies strongly from 1 pixel to 50 pixels or more.

For the lower clouds the object classification is performed in a similar way: optical thickness in the range $[0, 100]$ is divided into 10 intervals with a width of 10.



Next, a mean motion vector is calculated for each object and this vector is assigned to every pixel in the object. i.e. the object will move as a whole during the forecasting procedure. The forecast image produced this way is called object-based forecast. An example is shown in Fig. 3e and Fig. 3f for upper clouds. Fig. 3e shows the upper cloud optical thickness of the 13:15 UTC image used to produce the disparity vector field together with the corresponding object-based disparity vector field $\overrightarrow{V}_{obj}$ on top. Fig. 3f shows the object-based forecast of upper cloud optical thickness for a lead time of 1 h. One can observe that the object-based cloud motion vector field is much smoother and points mainly to the East in the southern part and to the North-East in the northern part. The front position is well captured by the object-based forecast when compared to the observation. Comparing the object-based forecast (Fig. 3f) to the pixel-based (Fig. 3d), it can be seen that the front line stays much more stable in the object-based forecast and the isolated cloud to the west (pixel position between 0 and 150 in x and between 150 and 200 in y) is moving as a whole and compares very well to the observation. However, e.g, the elongated cloud patches north of Spain (between pixel 100 and 150 in x and above pixel 200 in y) cannot be forecasted well and still the edge of the forecasted cloud layer looks too patchy.

### 3.2.3 Motion Vectors for Cloud Free Areas

As the motion vectors are derived from cloud optical thickness, the disparity vector field in the area between the clouds goes to zero (Fig. 3e). In case that cloud objects move into these regions, they stop. The thin line left of the front line and the squeezed cloud in the lower left corner (between pixel 50 and 150 in x and between 0 and 100 in y) in Fig. 3f show this effect. Through the mentioned advection of the disparity vector field before its use in the forecast this is compensated in part. In the following we illustrate that the remaining effect, for forecasts over extended lead times, is further minimized if the cloud free areas are filled with sensible motion estimates. The disparity vector field is divided into the object field for the clouds $\overrightarrow{V}_{obj}$ and the field $\overrightarrow{V}_{clr}$ for the clear sky areas called background. A weighted triangular interpolation of the disparity vector field between clouds is applied. A Delaunay triangulation creates a triangle mesh for interpolation between single cloud object related vectors returning a regular grid of interpolated values. Delaunay triangulations avoid sliver triangles by maximizing the minimum angle of all the angles in the triangulation. Therefore, a relatively uniform field can be created. Values inside the triangles issue from a smooth quintic interpolation of the windfield. The method used for the triangulation is the divide-and-conquer algorithm from Lee and Schachter (1980). In Figure 5 a weighted interpolation for the x-component of the disparity vector field is shown. In cloudy areas (black contour) the derived disparity vector field is used, while the interpolated values for the background field are calculated between the clouds (triangular shape). For forecast applications the values of the background field are limited to a range between -5 and 5 to avoid high gradients. The resulting disparity vector field (Fig. 3g) is significantly smoother than before (Fig. 3c).

### 3.3 Step 3: Intensity Correction for Quickly Thinning Convective Clouds

The pyramidal matcher (Sect. 2.3) can only predict the movement of the features in the images, i.e. rearrangement of values including divergence and convergence, but cannot create values in a given local area which could not be found, roughly speaking, within the "search radius" defined by the typical local wind/disparity vector (apart from the bilinear interpolation




implemented in Eq. 1). That means local development of values of optical thickness is very limited in our case since the matcher is rather thought to detect object displacement and distortion. In-situ formation of cirrus clouds, that are particularly important for DNI, cannot be predicted nor can the time and place of convective initiation. Once a cloud is observed its future evolution can be forecasted by continuation of the observed development: e.g., an increase in optical thickness in a cloud patch can be

forecasted through disparity vectors, as far as it can be represented by pure growth of areas with values of optical thickness present in the source image. Values larger than the ones found in the local surrounding around the cloud patch in the source image cannot be provided in the forecast. As mentioned before, the decaying stage of convective cells is of much interest for the purpose of DNI forecasting as thinning cirrus might allow an earlier recovery to DNI levels useful for CSP production (usually DNI > $200\,\mathrm{W\,m^{-2}}$). Opposed to the growing stage where, by no means, a nowcast of convective cell positions for

the future two hours is possible, for the decaying stage at least some useable initial information on the convective cloud is available. We found out that in this particular case, when convective cells start to decay, leaving behind a thinning anvil cirrus layer, the temporal evolution of the cloud optical thickness can be reasonably well forecasted or at least improved with respect to the output of the matcher.

To this end, quickly thinning convective clouds are first identified in satellite data and the successive evolution of their optical

thickness, as far as it can be forecasted through disparity vectors, is then corrected to follow typical temporal patterns. Both the identification of these clouds and the determination of typical values for the temporal evolution have been developed based on 300 cells detected by Cb-TRAM (stage 3, mature cells, according to the classification presented in Sect. 3.1.2). They have been investigated manually in an area covering Central and Southern Europe including the western part of Mediterranean Africa with a size of 1050×600 pixels (Fig. 6) and for the time period April-June 2013. Cells were classified as decaying cells in

cases where a decrease in optical thickness and a convergence of the anvil could be observed for the next time steps (temporal resolution 15 min). The "divergence" $div(\overrightarrow{V})$ is derived from the motion vector field $\overrightarrow{V} = (u,v)$ for each pixel:

$$div(\overrightarrow{V}) = (u_{right} - u_{left}) + (v_{above} - v_{below}) \tag{3}$$

with the motion vector components ($u$ and $v$) of the four neighbouring pixels above, below, right and left of the pixel under investigation: a positive value denotes a divergence, while a negative value indicates a convergence. Figure 7 shows the distri-

bution of the change in upper cloud layer optical thickness from one time step to the next averaged over an entire convective cell in relation to the average divergence of the given cell. The blue crosses denote cells which were found to be in decaying stage by eye and red ones for non-decaying cells. Obviously most blue crosses concentrate in a region with divergence smaller than -0.1 (horizontal line) and below a change in optical thickness of -0.01 (vertical line). As only a few red crosses for the apparently non-decaying cells lie in this area, these object averaged parameters can be used in an automated procedure for

identification of quickly thinning upper clouds:

– mean change in optical thickness from one time to the next is smaller than -0.01

– mean divergence $div$ (Eq. 3) of the motion vector field is smaller than -0.1.

Thus, a decrease in optical thickness and a slight converging movement indicate a decaying cell.





To determine a typical correction term for the temporal evolution of upper cloud optical thickness after the decaying phase has started, the subset of all 70 decaying cells has been investigated closer. An empirical modification derived from them is imposed onto the optical thickness of the convective objects forecasted through the disparity vectors. Before the application of disparity vectors as described in Sect. 3.2.2, optical thickness $\tau(P)$ of each pixel $P$ inside the convective object is decreased

by $f * \Delta\tau$. $f$ is the number of time steps after the forecast starts and $\Delta\tau$ is an empirical average optical thickness step found using the mentioned 70 cases:

$$\tau_{\mathrm{corr}}(P) = \tau(P) + \Delta\tau * f \,. \tag{4}$$

On its turn, the typical step $\Delta\tau$ is parameterised as a function of the observed mean optical thickness decrease $\Delta\tau_{\mathrm{initial}}$ of the convective object's optical thickness between the two initial images.

This information $\Delta\tau_{\mathrm{initial}}$ is selected because it depends on the convective cell under observation and because it is representative to the given atmospheric and physical conditions encountered. For application within the forecast procedure, the occurrence of $\Delta\tau/\Delta\tau_{\mathrm{initial}}$ in the range $[0,1]$ in bins of size 0.1 is investigated and shown in Figure 8 for a forecast of 15 min (left) and 45 min (right), where $\Delta\tau$ is the mean observed cell optical thickness decrease. It turns out that the mean initial optical thickness decrease of the convective cell is the strongest one and that the most typical decrease corresponds to half

this value for all forecast lead times up to 1 h. Therefore, forecasts for all decaying cells are implemented using a decrease $\Delta\tau = 0.5 * \Delta\tau_{\mathrm{initial}}$. This method is not reasonable for a forecast of more than one hour for the following reasons: 1) the remnants of the cells merge with other clouds and are not detectable any more; 2) the forecast and the observation of the cell differ strongly in shape, size and position. Thus, for a forecast of more than 1 h no further decrease in optical thickness is applied.

   One example of a decaying cell is shown in Fig. 9. The object-based forecast for 30 min without intensity correction (left)

predicts a larger ice optical thickness for the cell than it is in reality (middle). Fig. 9 (right) depicts the effect of the intensity correction. The upper cloud optical thickness predicted by application of this intensity correction is lower and more realistic than for the original forecast (left).

### 3.4   Step 4: Synthesis

After classification into upper and lower layer, object-based forecast of these layers and special correction for decaying upper

layer convective cirrus clouds, two fields of possibly overlapping forecast optical thickness are available. Fig. 3h shows, comparable to Fig. 3f, the 1-h forecast of upper cloud optical thickness and Fig. 3g the corresponding disparity vector field on top of the second initial image. By comparing both forecasts (Fig. 3f and Fig. 3h) the higher accuracy of the final forecast in terms of cloud coverage and optical thickness is clearly visible. In particular, the thin cloud patches in the northern part are better represented, the front is moving in a more compact way and the shape of cloud free regions in the South is more realistic.

However, the final forecast appears smoother than the observation because of the averaging procedure implemented for the cloud and cloud free objects and because of the interpolation procedure implemented in Eq. 1.



## 3.5 Step 5: Calculation of DNI

DNI computed in this paper considers only photons coming from the Sun that do not interact with the atmosphere (see the "strict definition" of DNI for numerical modeling of radiative transfer in Blanc et al. (2014)). In particular, no circumsolar radiation is taken into account. Thus,

$$DNI = \int E_0(\lambda) * exp(-\tau(\lambda)/\cos(sza))\, d\lambda\,, \tag{5}$$

according to Lambert-Beer's law. Here, the integral over wavelength $\lambda$ extends over the entire solar spectrum, $E_0(\lambda)$ represents the incoming solar radiation spectrum at top of atmosphere, $sza$ the solar zenith angle and $\tau(\lambda)$ the optical thickness of the atmosphere, including clouds, aerosols and (trace) gases. In the following, Eq. 5 is then approximated by means of

$$DNI = I_0 * exp(-(\tau_{aer} + \tau_g + \tau_{cld})/\cos(sza))\,, \tag{6}$$

with the broadband solar constant $I_0 = \int E_0(\lambda)d\lambda$. In Eq. 6, $\tau_{cld}$ represents the "broadband" cloud optical thickness, $\tau_{aer}$ the "broadband" aerosol optical thickness and $\tau_g$ the "broadband" optical thickness of the atmosphere, which mainly depends on water vapour (other trace gases, in particular ozone, play a minor role: ozone variability influences irradiance at the surface by « 1% (Lohmann et al., 2006)). Since only cloud optical properties are derived from MSG/SEVIRI, aerosol optical thickness $\tau_{aer}$ is set to zero, while the gas optical thickness $\tau_g$ has been computed for the mid-latitude summer standard atmosphere (Anderson et al., 1986). To this end, radiative transfer calculations with the radiative transfer model libRadtran (Mayer and Kylling, 2005; Emde et al., 2015) have been performed. Gas absorption is considered using the correlated-k approach by Kato et al. (1999) with 32 bands. The radiative transfer solver disort (Stamnes et al., 1988, 2000; Buras et al., 2011) is then applied to compute the direct irradiance at the surface $I_{sur}$. The gas optical thickness $\tau_g$ is computed from $I_{sur} = I_0 \exp(-\tau_g)$, where $I_0$ is the top of atmosphere irradiance. Different water vapour columns can be considered by modifying the total precipitable water in the radiative transfer model by rescaling the water vapour profile of the mid-latitude standard atmosphere. However, since no precipitable water is derived directly from MSG/SEVIRI, the precipitable water of $29.598\,\mathrm{kg\,m^{-2}}$ is used, which corresponds to $\tau_g = 0.292160$. For the term $\tau_{cld}$, a relationship between the "broadband" value and the cloud optical thickness at 550 nm $\tau_{cld,550\,nm}$ that is output of the cloud retrievals APICS and COCS is required. Again, radiative transfer calculations have been performed with the same settings as above but with the addition of a cloud layer of variable optical thickness. For liquid water clouds Mie optical properties are employed, for ice clouds the parameterisation by Baum et al. (2005) is used. For technical reasons, the radiative transfer computations are performed for the wavelength range 430–2060 nm. This includes 87% of the solar irradiance at top-of-atmosphere, and due to scattering and absorption, this figure is higher at the surface. Through a comparison between cloudy and cloud free surface irradiances the values for $\tau_{cld}$ are determined for both ice and liquid water clouds. It shows that $1 - \tau_{cld}/\tau_{cld,550\,nm}$ always remains well below 0.7% for water clouds and 0.1% for ice clouds such that this correction is neglected in the following.

Fig. 10 depicts the total cloud optical thickness provided by adding up lower and upper layer's values (left) for the same scene as in Fig. 3 and the computed DNI (right). The values range from $0\,W/m^2$ for areas with thick clouds (black) to around $900\,W/m^2$ for cloud free areas. Thin clouds reduce the DNI according to Eq. 6 as can be seen in the lower right corner.





## 4  Validation

For a validation of cloud and DNI forecasts two time periods, 4.3.2013 – 31.3.2013 and 1.7.2013 – 31.7.2013, were examined. These two months were chosen due to the appearance of different cloud types. The domain considered is the central part (marked in red) of the area investigated in Sect. 3.3 with a size of 751×501 pixels (Fig. 6). For March primarily advective

clouds are present in this domain with an increasing amount of convective clouds in July. During daytime (solar zenith angle < 80°) a forecast is started each full hour with a forecast horizon of 2 h and a time step of 5 min. Forecasts are compared to observations from MSG/SEVIRI. They represent the best result that a forecast algorithm based on such data can achieve. However, this is not an absolute validation of cloud cover and DNI, but takes into account that the forecast can only be as good as the input quantities are.

### 4.1  Cloud Masks

In order to quantitatively assess the performance of the forecast algorithm, we evaluate the capability of the algorithm to predict clouds and cloud free pixels by examining the errors of the forecast cloud mask against observed cloud masks from MSG/SEVIRI. Observations and forecast are connected through the contingency table (Table 3). Its four elements are the hits $a$, misses $c$, false alarms $b$ and correct negative events $d$. Hits represent the number of pixels that are correctly forecasted as

cloudy. Misses are the number of pixels that have been falsely predicted as cloud free although the observation is cloudy. False alarms are the number of pixels that are falsely predicted as cloudy although observations classify them as cloud free. And correct negatives are the number of pixels that are correctly forecasted as cloud free. Fig. 11 shows the four elements (hits in red, false alarms (fa) in green, misses in blue and correct negatives (cn) in white) for upper (left) and lower (right) cloud layers. Errors, especially for lower clouds, mainly occur at cloud edges or due to new developments or dissipating clouds.

The calculated parameters of the contingency table for all start times are averaged for every forecast time step up to 2 h (see beginning of Sect. 4 for the illustration of the forecast data set evaluated here). In Fig. 12 (left) the forecast errors (misses plus false alarms) for March (triangles) and July (crosses) are shown in percent with respect to the total number of pixels in the scene for upper (red) and lower (blue) clouds. For the upper cloud layer (but not for the lower clouds, see below) errors for persistence (the cloud distribution at the forecast starting point is assumed to stay unchanged) are plotted for comparison.

This comparison illustrates the benefit of the developed forecast algorithm. Forecasts errors are significantly lower compared to persistence with the smallest values for the 5 min forecasts (errors below 5% for upper cloud layer). At this time, persistence is still close to the observation, because clouds change only slightly during this time step. Afterwards, forecast errors increase smoothly with every time step to a maximum after 2 h. The difference between persistence and forecast increases also with time. For instance, for the upper cloud layer errors reached by persistence after about 1 h are reached by the forecast only after

2 h for March and July. Compared to persistence, the forecast at least doubles the lead time at a certain quality level. Most noticeable are the differences between the two cloud layers. The performance of the algorithm for the upper clouds shows much better results than for lower clouds. The reasons are: 1) the difficulty of a retrieval for lower clouds below thick upper clouds. This leads to errors in the forecast when not all clouds are detected in the initial images, a low cloud layer disappears





below a high one, or low clouds evolve into high clouds. 2) larger small scale variability for lower clouds, particularly in the case of convection. 3) formation of new lower clouds that cannot be forecasted.

Errors are smaller in July compared to March, because of the low cloud cover of 22.1% on average (62.7% in March) during this month (errors are relative to satellite scene size). For water clouds, detection and forecast are hindered by the presence of

upper clouds such that even a correct forecast might be incorrectly classified. Thus, it is difficult to assess the real accuracy of water cloud forecasts. For this reason, persistence for water clouds has not been evaluated.

In addition to the evaluation of the errors as shown above, we apply the Hanssen-Kuiper ($HK$) skill score (Hanssen and Kuipers, 1965) to our data set. This score has been widely used for the evaluation of meteorological fields since many years (Woodcock, 1976). It has been applied in particular to precipitation forecasts (e.g. III et al., 1990; Stephenson, 2000; Accadia

et al., 2003; Tartaglione, 2010; Gsella et al., 2014; Fekri and Yau, 2016) against observations but also cloud retrieval algorithms (Reuter et al., 2009; Bugliaro et al., 2011; Reuter and Fischer, 2014).

The Hanssen-Kuiper skill score (henceforth referred to as $HK$), also called Hanssen-Kuiper discriminant, Peirce skill score (Peirce, 1884), or true skill score (Flueck, 1987), combines the four elements of the contingency table (Table 3) in the following way:

$$HK = \frac{ad - bc}{(a+c)(b+d)}$$

$$= \frac{a}{a+c} + \frac{d}{b+d} - 1 \tag{7}$$

$$= \frac{a}{a+c} - \frac{b}{b+d} . \tag{8}$$

It can be expressed as the sum of the accuracy for events, i.e. the accuracy of forecasted clouds, (first term in Eq. 7, $\frac{a}{a+c}$, also called hit rate, $H$, or probability of detection, $POD$) and the accuracy for non-events, i.e. the accuracy of forecasted cloud free

pixels, (second term in Eq. 7, $\frac{d}{b+d}$). The subtraction of 1 in the end ensures that $-1 < HK < 1$. The $HK$ can also be expressed as the dfference between the hit rate $H = \frac{a}{a+c}$ (first term in Eq. 8) and the false alarm rate, $F = \frac{b}{b+d}$, or probability of false detection, $PODF$, (second term in Eq. 8). Thus, $HK$ is a measure of the hit rate relative to the false alarm rate and remains positive as long as $H$ is larger than $F$, i.e. indicates the ability of the forecast algorithm to produce correct cloud forecasts as well as to avoid false alarms. A skill score of 1 denotes a perfect match (all detected clouds have been forecasted, misses $c$ and

false alarms $b$ are zero), a score equal to -1 is related to a forecast not matching at all (hits $a$ and correct negatives $d$ are zero). Negative values are related to "inverse" forecasts and could be turned into positive values by interchanging forecasted events and non-events. A score of 0 is produced, e.g. by a forecast of a fully cloudy or fully cloud free scene, or by a "random" forecast, i.e. when $H$ and $F$ are equal. In this sense, the $HK$ represents the accuracy of the forecast in predicting the correct category with respect to the ability of a random selection. Furthermore, $HK$ is independent of the relative frequency of the observations

and also works with asymmetrical distributions, i.e. when more cloudy than cloud free pixels are present or viceversa. This is an important feature of this skill score since different geographical areas, different seasons and different times of the day are characterised by different cloud amounts that can vary considerably. In fact, if the cloud cover is low, i.e. if clear sky cover $(b+d)$ is large $(a+c \ll b+d)$, it is easy to correctly forecast $(d)$ the largest part of it, as errors can only arise from small





edge areas of small cloud cover (the forecast procedure does not account for cloud formation but rather shifts and modifies the shape of existing clouds). Thus, the second term in Eq. 7 is large, i.e. the second term in Eq. 8 is small (few false alarms). This tendency to large $HK$ score contributions due to high non-event accuracy (high accuracy of cloud free pixels) is balanced by the first term of the $HK$ skill score both in Eq. 7 and Eq. 8. There the same error potential (misplaced cloud edges) leads to

large cloud errors ($a$) compared to the small cloud cover ($a + b$). A low score contribution from events, i.e. a low hit rate, is the result. This way the direct effect of cloud cover on the skill score is minimised.

The resulting Hanssen-Kuiper ($HK$) skill score (Fig. 12, right) has been determined for upper (red) and lower (blue) clouds for both months with high values above 0.9 for the first time steps except for lower clouds in March (0.8) and a decrease to 0.55-0.7 after 2 h. As shown before the forecasts for upper clouds perform better than for lower clouds. The $HK$ skill score for

persistence for the upper clouds (black lines) is significantly lower than the respective forecasts (red lines) especially for July (black triangles vs. red triangles). Here, persistence already shows a lower skill for a 5 min forecast. This is mainly due to the lower hit rate of the persistence method with respect to the forecast in a situation where few clouds are present in the area under study (July has low cloud cover, see above): already a small displacement of the clouds can lead to significantly lower hit rates $H$ in this situation (see discussion of the $HK$ presented above). Differences of the accuracy of persistence for upper clouds

between March and July are evident: the two black lines in Fig. 12 (right) diverge with time, a hint that upper clouds forecasts for July are more difficult than for March due to the predominatly convective nature of clouds in July and to the inability of the matcher to forecast convective initiation and phase transition from liquid to solid during the convective process. However, the red curves in Fig. 12 (right) show that the $HK$ skill score of the upper cloud forecast becomes lower for July than for March for lead times larger than 40 min. The faster score decrease in July is due to the newly developing convective clouds

for which the forecast becomes inaccurate within a short period of time. The resulting larger cloud errors lead to lower values for July in the first term in Eq. 7 while the second term shows constantly high values due to the low cloud cover. The higher values in July for the first 40 min in the $HK$ skill score originate from the score contributions due to high cloud free pixels (non-events) accuracy (first term in Eq. 7). Nevertheless, the benefit in skill of the forecast compared to persistence can again be expressed as more than a doubling of lead times for a given score level. For the lower cloud layer (blue curves in Fig. 12,

right) the performance is better in July (blue triangles) than in March (blue crosses). This arises from the combination of a higher hit rate in March than in July, due to the larger lower cloud extent in March than in July and to the usually larger lower cloud sizes in March than in July (think of the frequent appearance of scattered cumuli in July) that makes it easier to forecast lower clouds in March, and of a higher false alarm rate in March than in July mainly due to the higher upper cloud coverage in March and the associated lower detection accuracy of lower clouds. The second effect outweighs the first one such that lower

cloud forecasts in July are more accurate than in March.

## 4.2 Cloud Optical Thickness

In order to test the performance of the algorithm with regard to the optical thickness a comparison of the forecasted optical thickness with the optical thickness observed from MSG/SEVIRI is done via a two dimensional histogram separated into upper (Fig. 13) and lower cloud layers (Fig. 14). The colorbar denotes the total number of occurrences. We selected the forecast





starting at 13:00 UTC on each day in March (Fig. 13 and Fig. 14 a,b,c) and July (Fig. 13 and Fig. 14 d,e,f) and compared forecast steps with the actual SEVIRI data measured. Comparisons of optical thickness for a 15 min, 1 h, and 2 h-forecast with observed optical thickness are shown separately in Fig. 13 for the higher and in Fig. 14 for the lower cloud layer.

For the upper cloud layer the algorithm shows an overall good performance with only small differences for most of the pixels for the 15min-forecast (Fig. 13a,d) and an increase of spread for the 1 h-forecast (Fig. 13b,e) and 2 h-forecast (Fig. 13c,f), which is illustrated by the 90% percentile (white contour line). The fact that the COCS algorithm produces results of either 0 or the range $[0.1 - 2.5]$ creates the narrow line without values between 0 and 0.1. Larger deviations mainly occur for observed or forecasted optical thickness equal to zero, where clouds have been forecasted but not observed and vice versa (false alarms and misses). Remarkable is the existence of two maxima in good agreement for small optical thickness around 0.25 and large values around 1.5. These reflect the essentially two types of high clouds mostly occuring: thin cirrus clouds and optical thick upper parts of deep clouds. The corresponding correlation coefficients are also shown and they confirm the good performance of the algorithm with high values over 0.9 for the upper clouds for the first forecast time steps. In analogy to the analysis of the $HK$ skill score, the correlation coefficients show higher values in March than in July, in particular for later time steps. This is most likely a consequence of the high frequency of convective clouds in July.

For the lower cloud layer (Fig. 14) the distribution is broader, due to the mentioned limitations of detection of lower clouds below higher ones. This is particularly true for March (Fig. 14a,b,c) and obvious in the missmatch for observed optical thickness at the largest values of optical thickness (forecasted and observed optical thickness of 100). This seems to be more difficult for March than for July, most likely because of multi-layered clouds around frontal systems.

The corresponding correlation coefficients show high values above 0.74 for the first time steps despite the lack in skill for cloud detection of lower clouds. Its deteriorating influence is apparent in the sharp decrease of the correlation coefficients for the 1 h- and 2 h-forecast.

To judge the quality of the forecast algorithm, histograms of the persistence method for the upper cloud layer are shown in Fig. 15. Compared to forecasts in Figure 13, deviation distributions are much broader and correlation coefficients significantly lower.

## 4.3 Direct Normal Irradiance

Fig. 16 illustrates the comparison of forecasted and observed Direct Normal Irradiance (DNI, Sect. 3.5) analogous to Fig. 13 and Fig. 14. DNI was calculated according to Eq. 6 with total cloud optical thickness from both cloud layers. Two maxima can be observed: 1) for low DNI in case of thick clouds and 2) for high DNI around $800\,W/m^2$ for March (Fig. 16a,b,c) and $900\,W/m^2$ for July (Fig. 16d,e,f) for cloud free cases with varying solar zenith angle. In analogy to the comparison of optical thickness, high deviations arise from cloud cover false alarms and misses and because of the difficulties of detecting multi-layer clouds. Remarkable is the sharply defined region at high DNI values ($> 600\ W/m^2$) showing a clear deficit of cases close to the 1:1 line. This is due to the fact that COCS does not provide measured values of optical thickness below 0.1. Thus large DNI values are missing in the "observation" derived, while the forecast can produce these optical thickness and DNI values.





The correlation coefficients for DNI are mostly higher than the values of both cloud types, especially for long forecasts, with lower values in July. This is due to the fact that forecasts are better for cloud areas with small optical thickness values than for optically thick clouds. Derivation of DNI emphasizes the relevance of these thin clouds for DNI predictions, while errors in the forecast of thick opaque clouds (e.g., new convective developments) are less detrimental.

## 5   Conclusions

Based on an optical flow method deriving cloud motion between two consecutive images, an algorithm for the forecast of cloud optical thickness and Direct Normal Irradiance has been developed for input data from the imager SEVIRI aboard the geostationary Meteosat Second Generation (MSG) satellite. The algorithms COCS and APICS provide cloud detection and cloud optical thickness for two vertically separated layers. Because of different velocities and motion directions these low and
high level clouds are forecasted separately for a time step of 5 min with a forecast horizon up to 2 h. To deal with the small scale fluctuations of the motion field derived for the two levels which would spoil forecasts of more than 15 or 30 min, an object classification is applied to the cloud layers and cloud-free background motion is interpolated. An intensity correction for decaying convective cells is implemented.

Using the observed cloud optical thickness as a reference, we could quantitatively validate the forecast for two months of
data March and July 2013. As far as cloud detection is concerned, the largest inaccuracy consists in the difficulty to retrieve clouds below optically thick clouds above. Consequently, forecast errors for the lower cloud layer are considerably higher than for high clouds. The forecast accuracy also differs for the two time periods because of different cloud coverage and cloud type. In March mainly fronts with many advective multi-layer clouds dominate in contrast to a high amount of shorter-lived convective clouds in July.

Convective clouds during July cause the forecast skill to decay quicker with forecast horizon than during March. For any given forecast quality requirement, over all cloud (or weather) types and for both cloud layers, a doubling of lead times was found comparing the developed forecast to a non-forecast, i.e., persistence.

The impact of weather situation also becomes apparent in the comparison of observed and forecasted optical thickness, which was carried out based on METEOSAT derived data. The distribution of deviations, analysed by means of 2D histograms, as
well as the correlation between forecast and observation show better results for March especially for a longer forecast. The wider scatter of deviations as well as lower correlation coefficients for comparison of optical thickness forecast and observation confirm the limitations of the forecast quality for low clouds compared to high clouds. Although much effort in this work was invested in separation of multi-layer clouds and their differential motion, still this stays a main source of uncertainty for satellite based nowcasting.

As final verification approach, direct normal irradiance DNI from forecasted optical thickness fields was compared to DNI derived from actual satellite observations. This approach is closest to an assessment of the forecast skill for real application in concentrating solar power generation. Most correct forecasts are, of course, found for the expected clear sky DNI and no direct irradiance below thick clouds. Still overall correlation even of 2 h forecast and observation derived values stays around 0.7.





As a next step comparison to ground-based irradiance measurements have to be conducted. However, this step would include not only the validation of the forecast but also of the DNI derivation at the surface. This step was only handled in a very simplified manner in this manuscript. At the moment the DNI is calculated with the optical thickness of trace gases and aerosols taken from standard atmospheres, but an implementation from other sources is feasible. E.g. water vapor from numerical

5  weather models and aerosol information from ground-based networks (e.g. AERONET) must be used to derive more precise surface irradiance.

An extension of the forecast horizon to 3 h or even more can be performed if needed. But as an increasing number of dissipating and newly developing clouds as well as the change in atmospheric motion would lead to a strong decrease in forecast accuracy these long-term forecasts should be treated with caution.

10  *Acknowledgements.* We acknowledge the European Commission for funding the project DNICast (www.dnicast-project.net), grant agreement 608623, and our colleague Hermann Mannstein, who passed away much too early in 2011, for a wealth of inventive and clever ideas that lead to these results, and for the development of the cloud matcher.



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



**Table 1.** SEVIRI spectral channels characteristics (Schmetz et al., 2002).

| Channel | $\lambda_{central}$ | $\lambda_{min}$ | $\lambda_{max}$ | Sampling Distance |
|---|---|---|---|---|
| | $\mu$m | $\mu$m | $\mu$m | km |
| VIS006 | 0.635 | 0.56 | 0.71 | 3 |
| VIS008 | 0.81 | 0.74 | 0.88 | 3 |
| IR_016 | 1.64 | 1.50 | 1.78 | 3 |
| IR_039 | 3.90 | 3.48 | 4.36 | 3 |
| WV_062 | 6.25 | 5.35 | 7.15 | 3 |
| WV_073 | 7.35 | 6.85 | 7.85 | 3 |
| IR_087 | 8.70 | 8.30 | 9.10 | 3 |
| IR_097 | 9.66 | 9.38 | 9.94 | 3 |
| IR_108 | 10.80 | 9.80 | 11.80 | 3 |
| IR_120 | 12.00 | 11.00 | 13.00 | 3 |
| IR_134 | 13.40 | 12.40 | 14.40 | 3 |
| HRV | Broadband (about 0.4–1.1) | | | 1 |



**Table 2.** Assignement of cloud optical thickness to two cloud classes called upper clouds and lower clouds. $\tau_{APICS,liq}$ is the APICS optical thickness for clouds with liquid cloud top phase, $\tau_{APICS,ice}$ is the APICS optical thickness for clouds with ice cloud top phase, $\tau_{COCS}$ is the COCS ice optical thickness, $\tau_{low}$ is the optical thickness assigned to the lower clouds, $\tau_{up}$ is the optical thickness assigned to the upper clouds.

| Liquid water cloud, no ice cloud above | | |
|---|---|---|
| $\tau_{APICS,liq} > 0 \wedge \tau_{COCS} = 0$ | $\longrightarrow \tau_{low} = \tau_{APICS,liq}$ | $\tau_{up} = 0$ |
| Thin cirrus cloud, no water cloud below | | |
| $\tau_{APICS,ice} \leq 2.5 \wedge \tau_{COCS} > 0$ | $\longrightarrow \tau_{low} = 0$ | $\tau_{up} = \tau_{COCS}$ |
| Thick cirrus cloud, no water cloud below | | |
| $\tau_{APICS,ice} > 2.5 \wedge (\tau_{APICS,ice} - \tau_{COCS}) \leq 2.3 \longrightarrow \tau_{low} = 0$ | | $\tau_{up} = \tau_{COCS}$ |
| Multi-layer cloud | | |
| $\tau_{APICS,ice} > 2.5 \wedge (\tau_{APICS,ice} - \tau_{COCS}) > 2.3 \longrightarrow \tau_{low} = \tau_{APICS,ice}$ | | $\tau_{up} = \tau_{COCS}$ |





**Table 3.** Contingency table.

| | | Observation | | |
|---|---|---|---|---|
| | Scenario | Cloudy | Cloud free | Total |
| Forecast | Cloudy | a | b | a+b |
| | Cloud free | c | d | c+d |
| | Total | a+c | b+d | N=a+b+c+d |



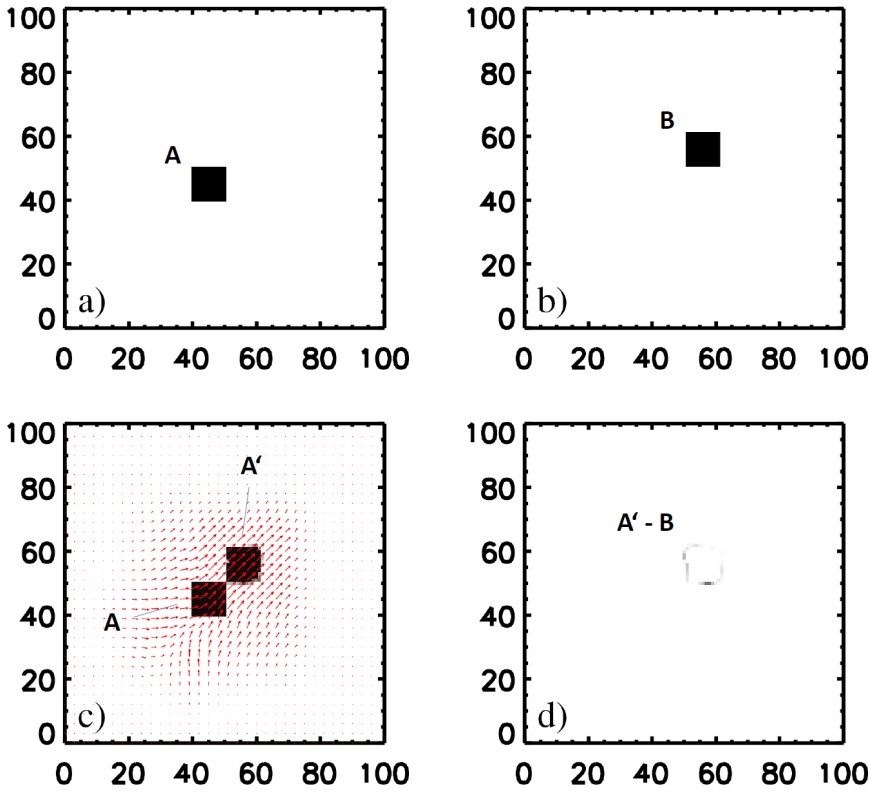

**Figure 1.** a) Start image $A$. b) Start image $B$. $A$ and $B$ are squares that have to be matched by the pyramidal matcher. c) The final disparity vector field $\overrightarrow{V}$ is plotted on the start image $A$ and $A'(P) = A(P - \overrightarrow{V}_{A \rightarrow B}(P))$ with d) the remaining difference field $A' - B$ after processing on all pyramid levels.





**Figure 2.** (a) False colour composite (VIS006, VIS008, IR_108) for 7 April 2013, 13:15 UTC, for the iberian peninsula. (b) The cloud mask for this scene with ice clouds in green, water clouds in blue, multi-phase clouds in red and cloud free areas in black and the optical thickness for upper (c) and lower (d) clouds.





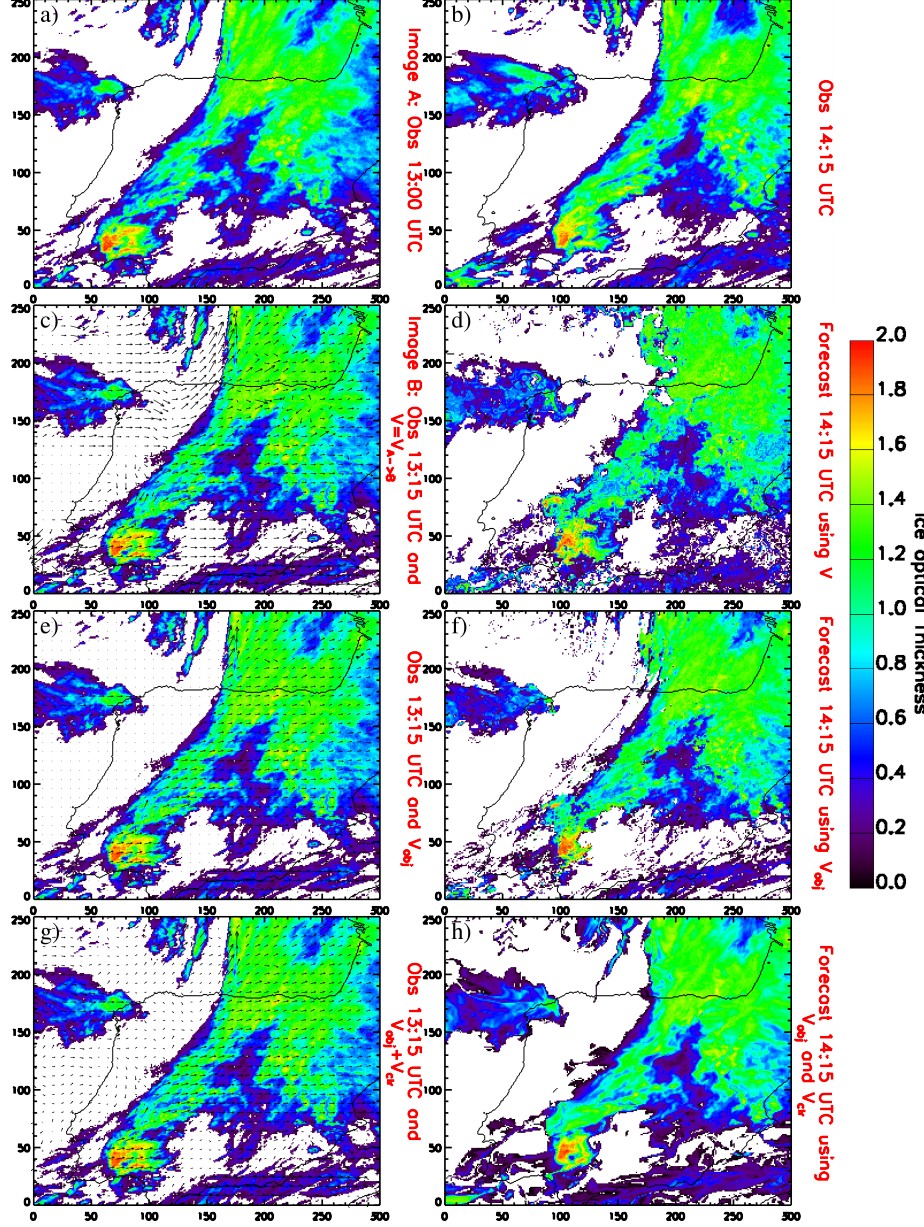

**Figure 3.** Illustration of the forecast of optical thickness for upper clouds for 7 April 2013 (a)+(c) Initial images A (13:00 UTC) and B (13:15 UTC) with the calculated pixel-based disparity vector field on top. (b) Upper cloud optical thickness at 14:15 UTC. (d) Pixel-based 1 h-forecast (i.e. for 14:15 UTC) of upper cloud optical thickness. (e) Upper cloud optical thickness at 13:15 UTC with the calculated object-based disparity vector field $\overrightarrow{V}_{obj}$ on top. (f) Object-based 1 h-forecast for 14:15 UTC of upper cloud optical thickness. (g) Upper cloud optical thickness at 13:15 UTC with the calculated object-based disparity vector field on top for cloudy areas $\overrightarrow{V}_{obj}$ and cloud free areas $\overrightarrow{V}_{clr}$. (h) Object-based 1 h-forecast for 14:15 UTC of upper cloud optical thickness including cloud free motion vectors.





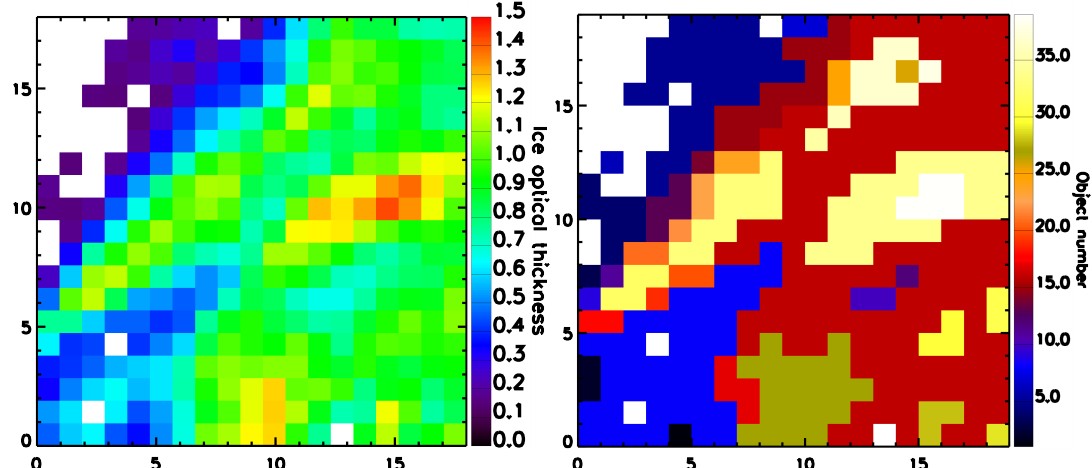

**Figure 4.** (Left) Upper cloud layer optical thickness extracted from the lower left part of Fig. 3c. (Right) Corresponding classification into 39 objects: Pixels with the same colour belong to the same object.





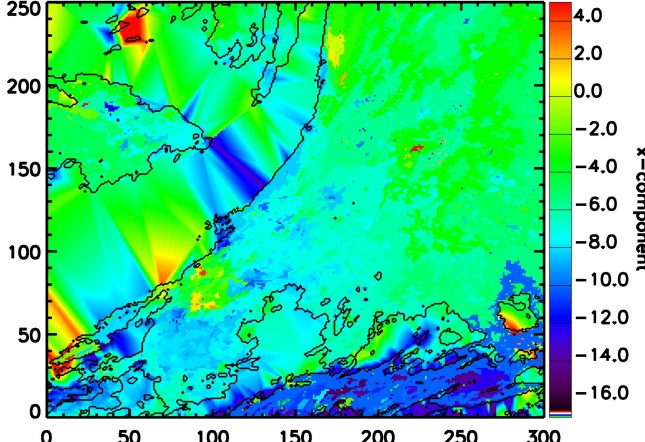

**Figure 5.** Delaunay triangulation for the x-component of the disparity vector field for upper clouds (black contours) returning a regular triangular grid of interpolated values between the clouds.





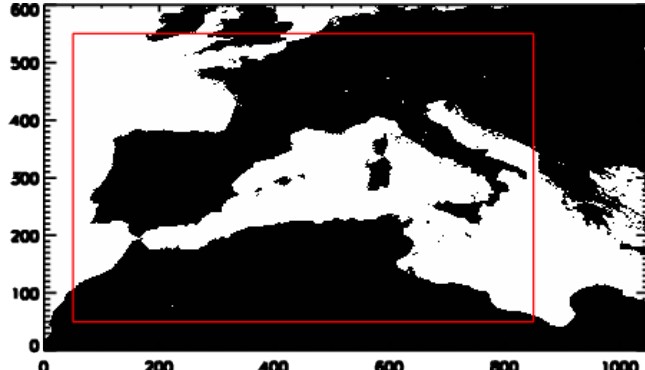

**Figure 6.** Domain used for the classification of decaying cells and for the validation presented in Sect. 4 (red square).





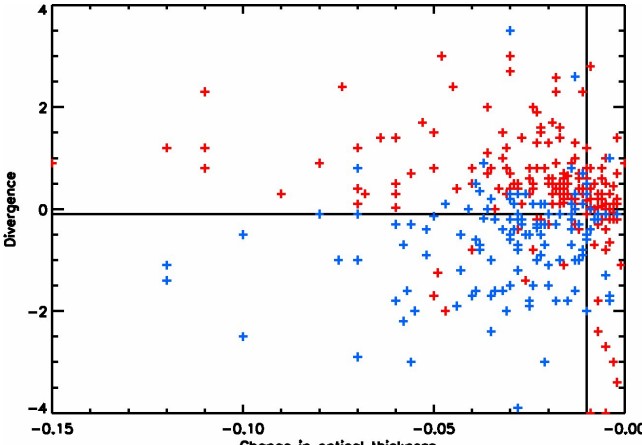

**Figure 7.** Distribution of the change in ice optical thickness in relation to the divergence with blue crosses denoting the decaying cells and red for the not decaying cells.




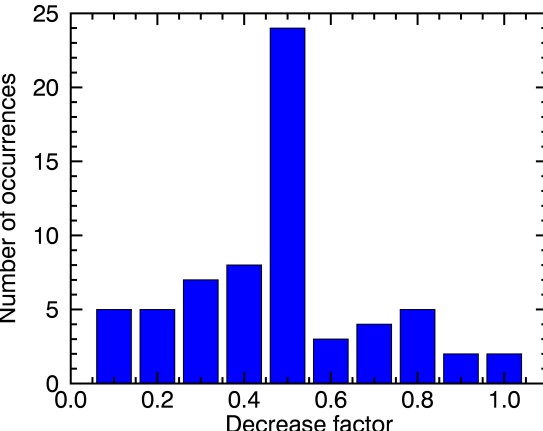
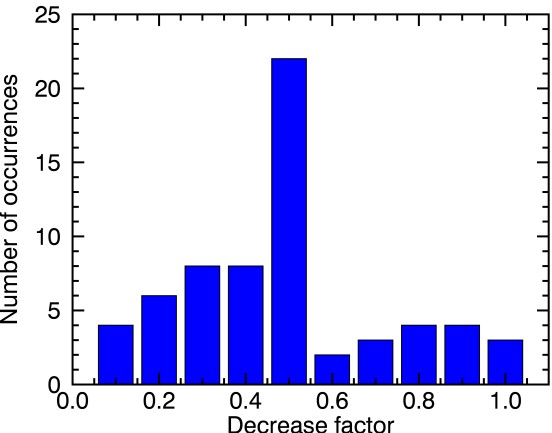

**Figure 8.** Distribution of the decrease factor $\Delta\tau/\Delta\tau_{\text{initial}}$ used to approximate the optical thickness decrease of a decaying cell (see text for details) for the 70 observed decaying cells for a forecast of 15 min (left) and 45 min (right).



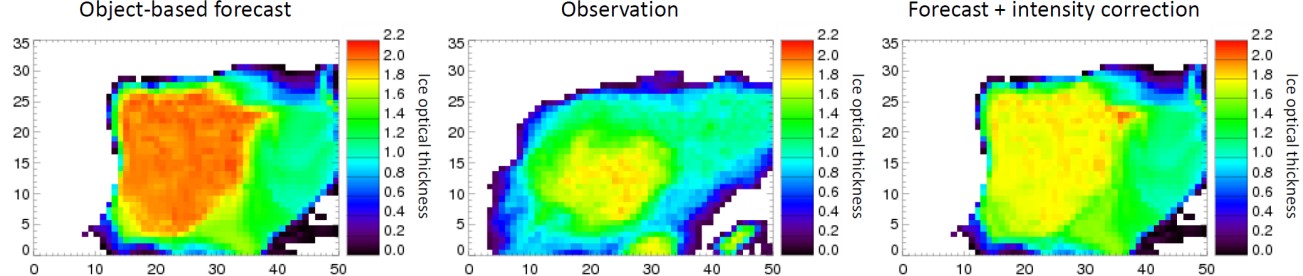

**Figure 9.** Upper cloud optical thickness for 9 June 2013, 17:00 UTC, for the real situation (middle) compared to the object-based forecast for 30 min (left) and the forecast with intensity correction (right).




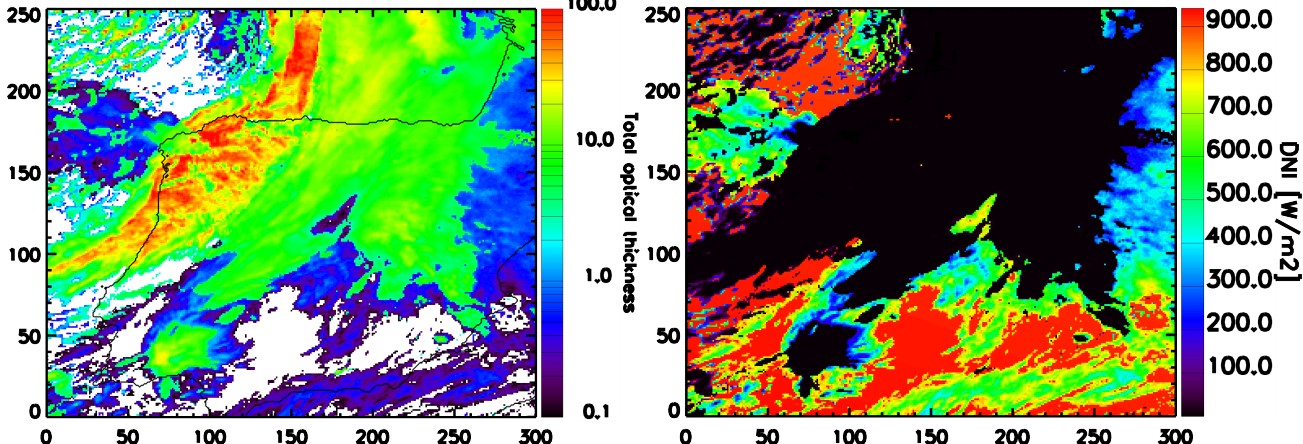

**Figure 10.** Optical thickness for upper and lower clouds together for 7 April 2013, 13:15 UTC (left) and the calculated direct normal irradiance in $W/m^2$ (right).





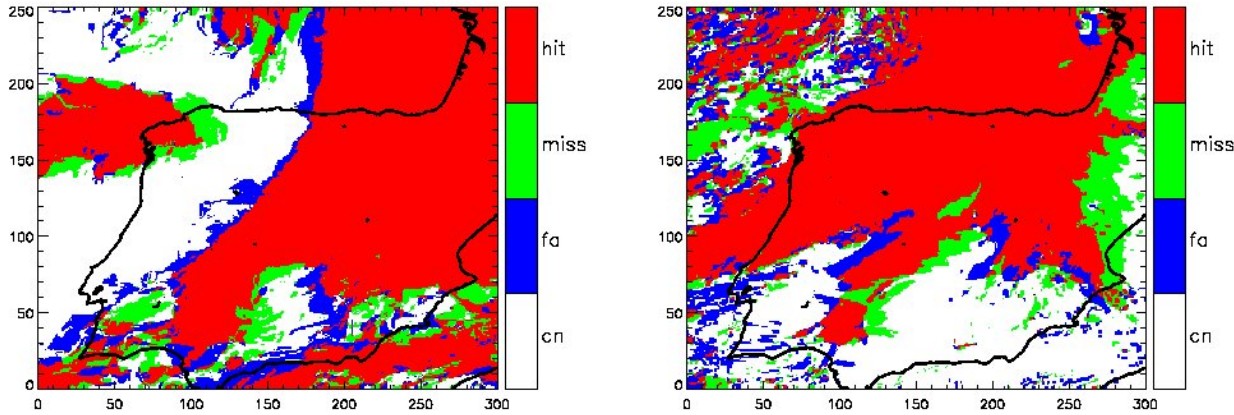

**Figure 11.** Illustration of the elements of the contingency table for upper (left) and lower (right) clouds with regard to the 1 h-forecast for 7 April 2013, 14:15 UTC: hits in red, false alarms (fa) in blue, misses in green and correct negatives (cn) in white.





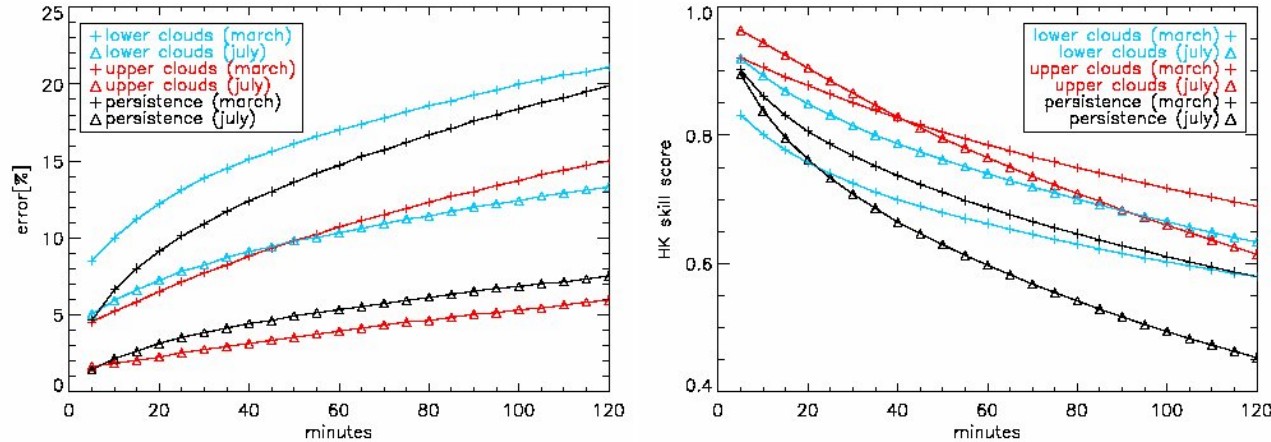

**Figure 12.** (a) Forecast errors (misses plus false alarms) and (b) the Hanssen-Kuiper skill score for March (triangles) and July (crosses) in percent for upper (red) and lower (blue) cloud layers and persistence for upper clouds (black).





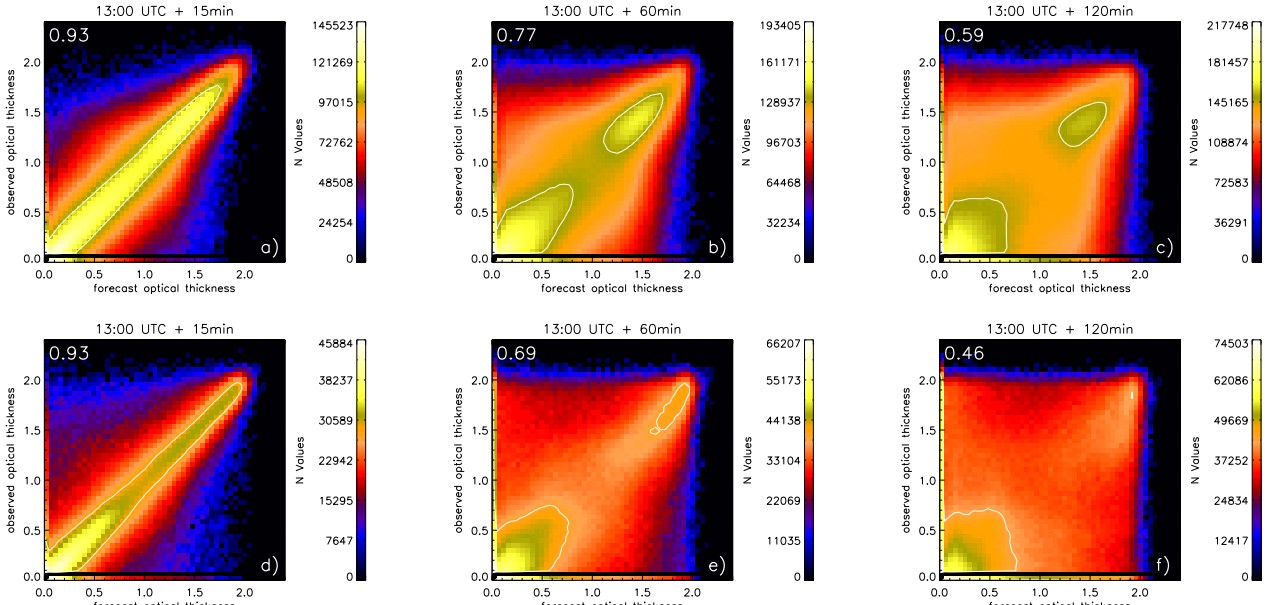

**Figure 13.** 2D histogram of the forecasted optical thickness of the upper cloud layer compared to the real optical thickness with forecasts starting at 13:00 UTC every day: for a 15min-forecast (a) 1 h-forecast (b) and 2 h-forecast (c) in March and July respectively (d,e,f). Colours denote the total number of occurrences. The number in the upper left corner of all images is the correlation coefficient. The white contour line denote the 90% percentile.





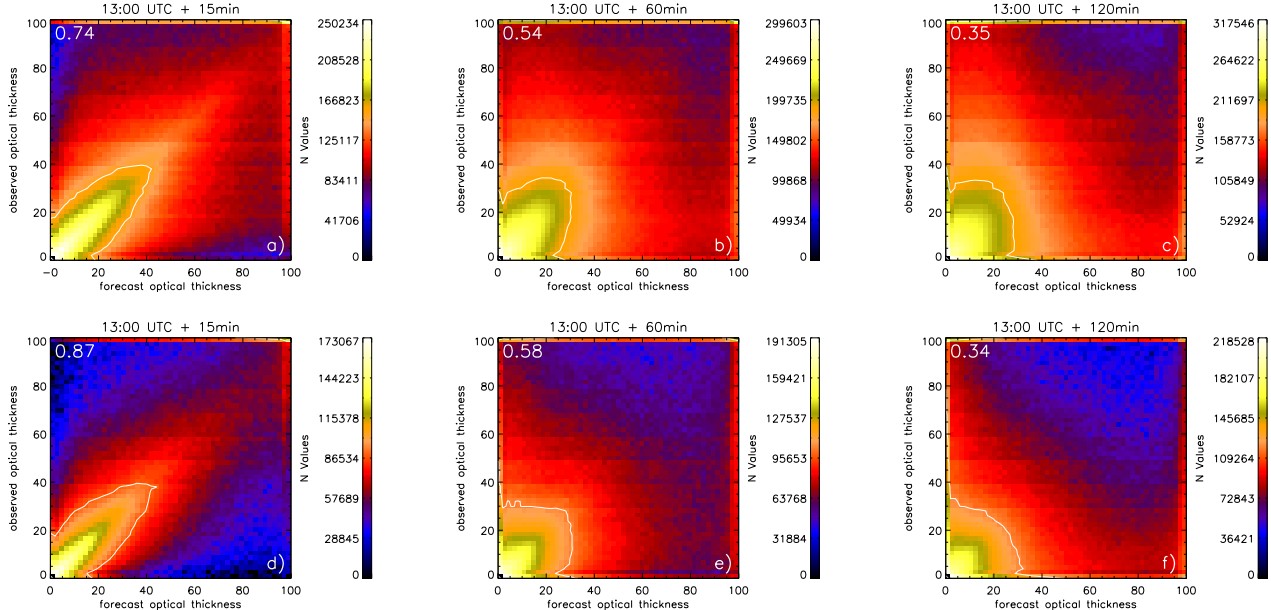

**Figure 14.** 2D histogram of the forecasted optical thickness for the lower cloud layer compared to the real optical thickness with forecasts starting at 13:00 UTC every day: for a 15min-forecast (a), 1 h-forecast (b) and 2 h-forecast (c) in March and July respectively (d,e,f). Colours denote the total number of occurences. The number in the upper left corner of all images is the correlation coefficient. The white contour line denote the 90% percentile.





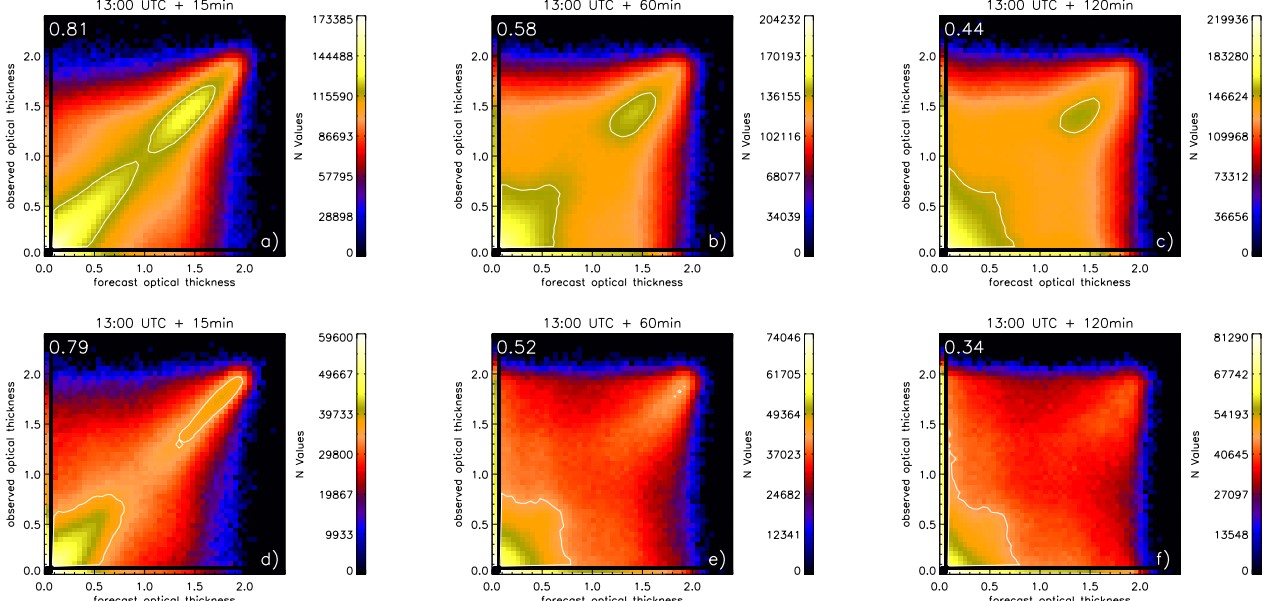

**Figure 15.** 2D histogram of the observed optical thickness of the upper cloud layer compared to the persistence optical thickness for a time difference of 15 min (a), 1 h (b) and 2 h (c) in March and July respectively (d,e,f). Colours denote the total number of occurrences. The white contour line denote the 90% percentile.





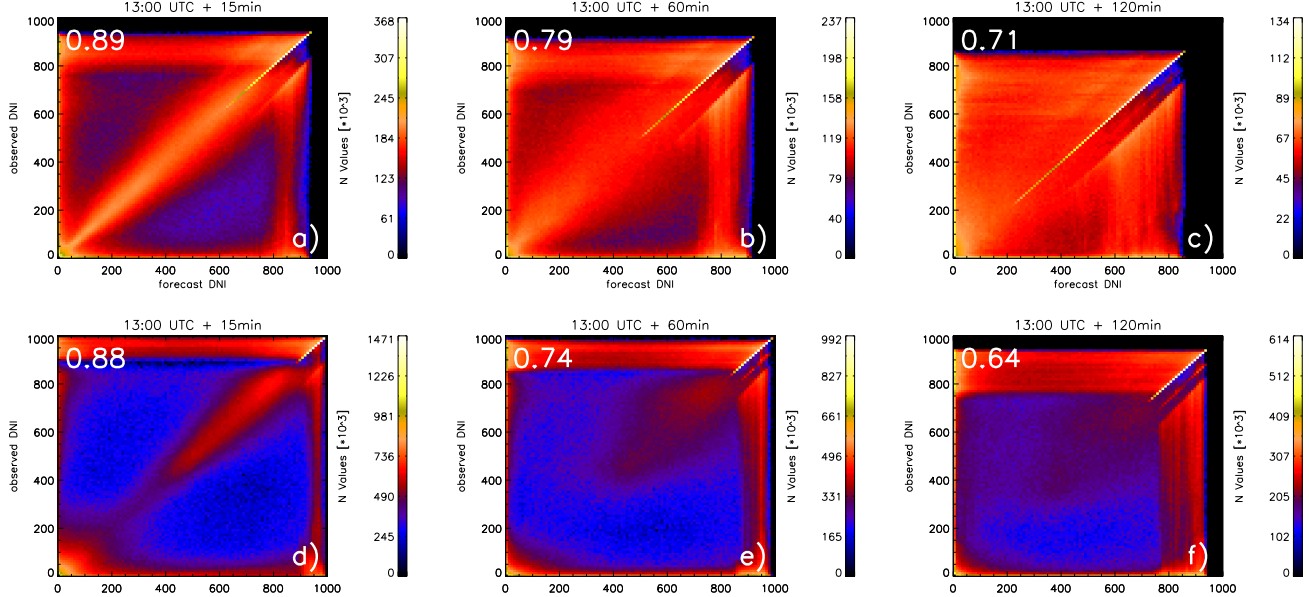

**Figure 16.** 2D histogram of the forecasted DNI compared to the observed DNI with forecasts starting at 13:00 UTC every day: for a 15min-forecast (a), 1 h-forecast (b) and 2 h-forecast (c) in March and July respectively (d,e,f). Colours denote the total number of occurrences.