# Peer review of "Cloud and DNI nowcasting with MSG/SEVIRI for the optimised operation of concentrating solar power plants"

_Atmospheric Measurement Techniques, 2016_

## Referee Comment (RC1) · Anonymous Referee #2 · 12 Oct 2016

This manuscript reports the algorithm for nowcasting direct normal irradiance based on satellite measurements of cloud properties and movements. This has great significance for solar/gas power plant in arranging power-generation mode change and has significant economical value for stability of the power supply and for saving gas from loss during transition period. The manuscript is well written, the algorithm is reasonable, results seem flawless, and conclusion is meaningful. This reviewer recommends this manuscript be published with only minor optional revisions.

1. The paper is too long. No people today have time to read so long scientific article. The authors should try their best to make it concise. Significant reduction of the pages is mandatory.

[Figure]

2. In Section 2.1, the authors stated that COCS can detect 50% of the cirrus clouds with OT ∼0.1. How this 50% number is obtained? During daytime, even CALIPSO lidar cannot claim this due to sunlight noise, let alone to say the narrow field of view of lidar, which covers negligible area of the earth.

3. Section 2.3 is not clearly presented. Find a better way to present for this section.

4. Organization is loose for the whole manuscript. Merge some sections and consider a better sequence for each section.

---

## Author Comment (AC1) · 13 Oct 2016

We thank the reviewer for his positive evaluation of our manuscript. However, Point 2 and 3 have already been clarified after the quick Review:

* In Section 2.1, the authors stated that COCS can detect 50% of the cirrus clouds with OT $\sim$0.1. How this 50% number is obtained? During daytime, even CALIPSO lidar cannot claim this due to sunlight noise, let alone to say the narrow field of view of lidar, which covers negligible area of the earth.

This is true. COCS detects 50% of the cirrus clouds with OT$\sim$0.1 "observed by CALIPSO". This has been clarified in the text.

[Figure]

\* Section 2.3 is not clearly presented. Find a better way to present for this section.

Section 2.3 has been rephrased. The explanation of the pyramidal matcher is done now step by step in a clearer way. Please refer to the manuscript for details.

Regarding Point 1 and 4, several paragraphs have been slightly shortened and modified.

Nevertheless, we refrain from making strong changes to the manuscript until we have concrete suggestions from the reviewers about which sections or paragraphs should be modified or clarified or shortened without loss of information for the reader. We would be grateful if reviewer #2 could consider the changes that have already been made for the current review process and if he could indicate which sections necessitate, to his opinion, additional clarifications and which ones should be merged/shortened.

---

## Referee Comment (RC2) · Anonymous Referee #3 · 2 Dec 2016

Review: Cloud and DNI nowcasting with MSG/SEVIRI for the optimized operation of concentrating solar power plants. By: T. Sirch, L. Bugliaro, T. Zinner, M. Möhrlein and M. Vazquez-Navarro

General:

The authors are presenting an interesting approach to improve the nowcasting of clouds and direct normal irradiance for solar power systems by using sequential satellite images from Meteosat SEVIRI. Many different uncertainties are discussed, including the evolution and differential motion of cirrus clouds, convective clouds and multi-layer clouds. The study explicitly describes different approaches to estimate these cloud motions. The manuscript is nicely written and the figures support the forecast al-

gorithm in a very illustrative way. Some paragraphs are still a bit confusing and require slight improvements, which are mentioned in the following. In my opinion this paper deals with a very important topic and should be published in AMT, however, I do have some questions and comments.

Specific comments:

(1) The manuscript contains many detailed descriptions about the different approaches, which are important to understand the method. The sections about the tools and forecast algorithms are, however, very long and include some equations (e.g. page 11), which are not necessary for understanding. I believe that the manuscript can be substantially improved if the authors reshape paragraphs and shorten some of the explanations. I instead miss an explicit outlook, that discusses the applicability of the presented algorithms for operational cloud forecasts for solar power systems. What might be a limit for a reliable DNI forecast? You mention that a three hour forecast might be possible, if needed. It would be very helpful to provide some additional information about what is required by the energy companies.

(2) Besides differential motions of multi-layer clouds (which are described in detail), broken cumulus cloud fields remain the biggest source of uncertainty for DNI forecasts due to their sub-pixel inhomogeneity and short life times. From the convective clouds point of view, only the influence of quickly thinning clouds is discussed. Cloud optical thickness forecast is also strongly influenced by this sub-pixel inhomogeneity (generally underestimated due to plane parallel albedo bias), albeit cannot be resolved by Meteosat (see Wolters et al. 2010: Broken and inhomogeneous cloud impact on satellite cloud particle effective radius and cloud-phase retrievals; Koren et al. 2008: How small is a small cloud?). These effects should be better acknowledged in the present study. Also rapidly changing small-scale convective cloud fields with very short time scales but low advection speeds (see Bley et al. 2016: Meteosat-Based Characterization of the Spatiotemporal Evolution of Warm Convective Cloud Fields over Central Europe) can influence the accuracy of the DNI forecast. The authors don't have to

quantify these broken cloud effects explicitly, but at least mention the importance of these effects in the conclusions, as briefly done in the introduction (short forecasts with Total Sky Imagers)

Minor technical comments:

(1) Page 1, Line 11: Rephrase the sentence with double "towards". It is not clear, if the 80% is still achieved after 2 h or less.

(2) Page 1, Line 15: Remove "at least".

(3) Page 2, Line 9: You introduce the persistence model and should describe it with one sentence (You are not doing this until Page 17, Line 24). Further change the sentence to: "of low cloud variability and obviously for clear-sky cases".

(4) Page 2, Line 14: "," is missing between "5-6 h" and "forecasting methods"

(5) Page 3, Line 10-12: Split into two sentences.

(6) Page 3, Line 22: Only use the abbreviation DNI, after you introduced it.

(7) Page 5, Line 2 : Which methods are used from Cb-TRAM? If it is the whole algorithm, than rather remove "methods of".

(8) Page 5, Line 20: Comma missing after "patterns".

(9) Page 6, Line 20: "tmporal" should be temporal

(10) Page 6, Line 26: "Notice, that the it is different", seems to be incorrect.

(11) Page 7, Line 12: In the following,

(12) Page 7, Line 31: For a slant optical thickness of 7. You could add a reference here.

(13) Page 8, Line 5-11: Double description of the yellow and blue colors. You can describe it in the caption of the figure and than just refer to it in line 9-11.

[Figure]

(14) Page 8, Line 11: "sngle layer", should be single.

(15) Page 9, Line 4: You mention that optical thick clouds are not important with respect to CSP production. I guess this is only valid for homogeneous clouds. What happens for optical thick cumulus cloud fields with a lot of cloud free gaps in between?

(16) Page 10, Line 23-24: rephrase "the area used" to "the selected area"

(17): Page 12, Line 5: "de" should be changed to "be".

(18): Page 12, Line 10: I suggest to write: "Nonetheless, the cloud motion vectors still exhibit a strong horizontal variability, especially..."

(19): Page 13, Line 3-5: Figure description can be removed, as it is already in the caption.

(20): Page 13, Line 17: "partially compensated" sound better.

(21): Page 13, Line 25: "In Figure 5, a" The comma is important here.

(22): Page 14, Line 1: Change to "This means that local..."

(23) Page 18, Lines 1-6: Similar to major comment (2), there are more reasons for high uncertainties in case of low cumulus clouds. Please include somewhere in the paper, how the sub-pixel variability can influence the optical thickness accuracy (Wolters et al. 2010, Koren et al. 2008) and estimation of cloud life time (Bley et al. 2016, JAMC) which lead to uncertainties for solar power applications.

(24) Page 19, Line 15: "upper clouds forecasts" needs to be changed to "upper cloud forecasts"

(25) Page 19, Line 16: Spelling mistake in "predominatly"

(26) Page 19, Line 25 ff. "This arises..." Very long sentence. Splitting into two sentences would make it much clearer.

(27) Page 20, Line 17: Large optical thickness values of 100 should be interpreted with

caution, because of saturation effects in the satellite retrieval of the optical thickness, which basically set the maximum limit to 100.

(28) Page 21, Line 3: What about 3D radiative effects of small convective clouds, which likely cause high uncertainties in the DNI predictions?

(29) Page 21, Line 18: Shouldn't it be "cloud types"?

(30) Page 21, Line 31: Spelling mistake. I suggest: "for a real application" or "for real applications".

(31) Page 21, Line 33: Please improve the structure of the last sentence. Suggestions: "However, the overall correlation between the 2 h forecast and the observation is still around 0.7

(32) Page 22, Line 1: Spelling mistake. Change to "comparisons".

(33) Page 28, Table.2: "Assignement" is misspelled.

---

## Author Comment (AC2) · 22 Dec 2016

We thank the reviewer for his positive evaluation of our manuscript. The reviewer questions and/or comments are answered in the following.

1) The paper is too long. No people today have time to read so long scientific article. The authors should try their best to make it concise.

The manuscript has been shortened by 1.5 pages in total. Following the suggestions of the reviewers, paragraphs and explanations have been reshaped/shortened in the following sections: 2.1, 2.2, 3.1.1, 3.2.1, 3.2.2, 3.5 and 5. Please refer to the manuscript for details.

2) In Section 2.1, the authors stated that COCS can detect 50% of the cirrus clouds with OT $\sim$0.1. How this 50% number is obtained? During daytime, even CALIPSO lidar cannot claim this due to sunlight noise, let alone to say the narrow field of view of lidar, which covers negligible area of the earth.

This is true. COCS detects 50% of the cirrus clouds with OT$\sim$0.1 "observed by CALIPSO". This has been clarified in the text.

3) Section 2.3 is not clearly presented. Find a better way to present for this section.

Section 2.3 has been rephrased. The explanation of the pyramidal matcher is done now step by step in a clearer way. Please refer to the manuscript for details.

4) Organization is loose for the whole manuscript. Merge some sections and consider a better sequence for each section.

We think the organisation of the paper is reasonable but maybe the line of thought is partly obscured by the wealth of contents presented, so we highlight it again:

a) Introduction (Section 1)

b) The tools used for the handling of the satellite data to extract information about cloud location, cloud phase and cloud type are described in Section 2 together with the satellite instrument MSG/SEVIRI.

c) The forecast algorithm is presented step by step in Section 3:

step 1: cloud classification (Section 3.1)

step 2: derivation of the motion vector field (Section 3.2)

step 3: intensity correction for convective cells (Section 3.3)

step 4: final forecast (Section 3.4)

step 5: calculation of DNI from the forecasted cloud optical properties (Section 3.5)

This section is designed such that the reader should be able to re-implement the forecast algorithm in a straightforward way.

d) In Section 4 the validation for the cloud masks (Section 4.1), the cloud optical thickness (Section 4.2) and DNI (Section 4.3) is shown.

e) Conclusions are in Section 5.

Furthermore, we improved the motivation in Section 3.1.1 and modified the sequence of the argumentation.
* * *

---

## Author Comment (AC3) · 22 Dec 2016

We thank the reviewer for his positive evaluation of our manuscript and for his constructive comments that helped improving the paper. The reviewer questions and/or comments are answered in the following.

Specific comments:

(1) The manuscript contains many detailed descriptions about the different approaches, which are important to understand the method. The sections about the tools and forecast algorithms are, however, very long and include some equations (e.g. page 11), which are not necessary for understanding. I believe that the manuscript can be substantially improved if the authors reshape paragraphs and shorten some of the explanations. I instead miss an explicit outlook, that discusses the applicability of the presented algorithms for operational cloud forecasts for solar power systems. What might be a limit for a reliable DNI forecast? You mention that a three hour forecast might be possible, if needed. It would be very helpful to provide some additional information about what is required by the energy companies.

The manuscript has been shortened by 1.5 pages in total. Following the suggestions of the reviewers, paragraphs and explanations have been reshaped/shortened in the following sections: 2.1, 2.2, 3.1.1, 3.2.1, 3.2.2, 3.5 and 5. In the conclusion a paragraph has been added providing some additional information about what is required by solar power plant operators. Please refer to the manuscript for details.

(2) Besides differential motions of multi-layer clouds (which are described in detail), broken cumulus cloud fields remain the biggest source of uncertainty for DNI forecasts due to their sub-pixel inhomogeneity and short life times. From the convective clouds point of view, only the influence of quickly thinning clouds is discussed. Cloud optical thickness forecast is also strongly influenced by this sub-pixel inhomogeneity (generally underestimated due to plane parallel albedo bias), albeit cannot be resolved by Meteosat (see Wolters et al. 2010: Broken and inhomogeneous cloud impact on satellite cloud particle effective radius and cloud-phase retrievals; Koren et al. 2008: How small is a small cloud?). These effects should be better acknowledged in the present study. Also rapidly changing small-scale convective cloud fields with very short time scales but low advection speeds (see Bley et al. 2016: Meteosat-Based Characterization of the Spatiotemporal Evolution of Warm Convective Cloud Fields over Central Europe) can influence the accuracy of the DNI forecast. The authors don't have to quantify these broken cloud effects explicitly, but at least mention the importance of these effects in the conclusions, as briefly done in the introduction (short forecasts with Total Sky Imagers).

The influence of sub-pixel inhomogeneity by broken cumulus cloud fields and of rapidly

changing small-scale convective cloud fields on the accuracy of DNI forecasts has been acknowledged in the manuscript. In section 4 and 5 the importance of these effects has been mentioned together with the recommended references.

Minor technical comments:

(1) Page 1, Line 11: Rephrase the sentence with double "towards". It is not clear, if the 80% is still achieved after 2 h or less.

The 80% is achieved after 2h. This has been clarified in the text.

(2) Page 1, Line 15: Remove "at least".

Removed.

(3) Page 2, Line 9: You introduce the persistence model and should describe it with one sentence (You are not doing this until Page 17, Line 24). Further change the sentence to: "of low cloud variability and obviously for clear-sky cases".

The sentence has been changed.

(4) Page 2, Line 14: "," is missing between "5-6 h" and "forecasting methods"

Implemented.

(5) Page 3, Line 10-12: Split into two sentences.

The sentence has been split into two.

(6) Page 3, Line 22: Only use the abbreviation DNI, after you introduced it.

Implemented.

(7) Page 5, Line 2: Which methods are used from Cb-TRAM? If it is the whole algorithm, than rather remove "methods of".

Only methods of Cb-TRAM are used, not the whole algorithm. This has been clarified in the text.

(8) Page 5, Line 20: Comma missing after "patterns".

Implemented.

(9) Page 6, Line 20: "tmporal" should be temporal

Changed .

(10) Page 6, Line 26: "Notice, that the it is different", seems to be incorrect.

This is true, it has been clarified in the text.

(11) Page 7, Line 12: In the following,

Implemented.

(12) Page 7, Line 31: For a slant optical thickness of 7. You could add a reference here.

We used radiative transfer computations. This has been clarified in the text.

(13) Page 8, Line 5-11: Double description of the yellow and blue colors. You can describe it in the caption of the figure and than just refer to it in line 9-11.

One description has been removed.

(14) Page 8, Line 11: "sngle layer", should be single.

Changed.

(15) Page 9, Line 4: You mention that optical thick clouds are not important with respect to CSP production. I guess this is only valid for homogeneous clouds. What happens for optical thick cumulus cloud fields with a lot of cloud free gaps in between?

Optical thick clouds are important for DNI assessment since they reduce DNI to zero. However, and this is what we meant in the manuscript, exact knowledge of the optical thickness of thick clouds in general is not important with respect to CSP production since both e.g. an optical thickness of 40 and 80 have the same effect on DNI.

(16) Page 10, Line 23-24: rephrase "the area used" to "the selected area"

Rephrased.

(17) Page 12, Line 5: "de" should be changed to "be".

Changed.

(18) Page 12, Line 10: I suggest to write: "Nonetheless, the cloud motion vectors still exhibit a strong horizontal variability, especially..."

Implemented.

(19) Page 13, Line 3-5: Figure description can be removed, as it is already in the caption.

Removed.

(20) Page 13, Line 17: "partially compensated" sound better.

Changed.

(21) Page 13, Line 25: "In Figure 5, a" The comma is important here.

Implemented.

(22) Page 14, Line 1: Change to "This means that local..."

Changed.

(23) Page 18, Lines 1-6: Similar to major comment (2), there are more reasons for high uncertainties in case of low cumulus clouds. Please include somewhere in the paper, how the sub-pixel variability can influence the optical thickness accuracy (Wolters et al. 2010, Koren et al. 2008) and estimation of cloud life time (Bley et al. 2016, JAMC) which lead to uncertainties for solar power applications.

The described influences have been implemented in Sections 4 and 5 together with the recommended references. Please refer to the manuscript for details.

[Figure]

(24) Page 19, Line 15: "upper clouds forecasts" needs to be changed to "upper cloud forecasts"

Changed.

(25) Page 19, Line 16: Spelling mistake in "predominatly"

Changed.

(26) Page 19, Line 25 ff. "This arises..." Very long sentence. Splitting into two sentences would make it much clearer.

The sentence has been split into two.

(27) Page 20, Line 17: Large optical thickness values of 100 should be interpreted with caution, because of saturation effects in the satellite retrieval of the optical thickness, which basically set the maximum limit to 100.

This is true. It has been clarified in the text.

(28) Page 21, Line 3: What about 3D radiative effects of small convective clouds, which likely cause high uncertainties in the DNI predictions?

By definition DNI computed in this paper considers only photons coming from the sun that do not interact with the atmosphere (see section 3.5). Also the contribution of 3D radiative effects to DNI is supposed to be negligible.

(29) Page 21, Line 18: Shouldn't it be "cloud types"?

Yes, this has been changed.

(30) Page 21, Line 31: Spelling mistake. I suggest: "for a real application" or "for real applications".

Changed.

(31) Page 21, Line 33: Please improve the structure of the last sentence. Suggestions:

"However, the overall correlation between the 2 h forecast and the observation is still around 0.7

Changed.

(32) Page 22, Line 1: Spelling mistake. Change to "comparisons".

Changed.

(33) Page 28, Table.2: "Assignement" is misspelled.

Changed.
* * *